# How steady are steady-state mountain belts? – a re-examination of the Olympic Mountains (Washington State, USA)

Lorenz Michel[1], Christoph Glotzbach[1], Sarah Falkowski[1], Byron A. Adams[1,2], Todd A. Ehlers[1]

[1]Department of Geosciences, University of Tübingen, Tübingen, 72074, Germany
[2]School of Earth Sciences, University of Bristol, Bristol, BS8 1RJ, United Kingdom

*Correspondence to*: Todd A. Ehlers (todd.ehlers@uni-tuebingen.de)

**Abstract.** The Olympic Mountains of Washington State (USA) represent the aerially exposed accretionary wedge of the Cascadia Subduction Zone and are thought to be in flux steady-state, whereby the mass outflux (denudation) and influx (tectonic accretion) into the mountain range are balanced. We use a multi-method approach to investigate how temporal variations in the influx and outflux could affect previous interpretations of flux steady-state. This includes analysis of published and new thermochronometric ages for (U-Th)/He dating of apatite and zircon (AHe and ZHe, respectively) and fission track dating of apatite and zircon (AFT and ZFT, respectively), 1D thermo-kinematic modelling of thermochronometric data, and independent estimates of outflux and influx.

In total, we present 61 new AHe, ZHe, AFT, and ZFT thermochronometric ages from 21 new samples. AHe ages are generally young (<4 Ma), and, in some samples, AFT ages (5–8 Ma) overlap with ZHe ages (7–9 Ma) within uncertainties. Thermo-kinematic modelling shows that exhumation rates are temporally variable, with rates decreasing from >2 km/Myr to <0.3 km/Myr around 5–7 Ma. With the onset of Plio-Pleistocene glaciation, exhumation rates increased to values >1 km/Myr. This demonstrates that the material outflux is varying through time, requiring a commensurate variation in influx to maintain flux steady-state. Evaluation of the offshore and onshore sediment record shows that the material influx is also variable through time and that the amount of accreted sediment in the wedge is spatially variable. This qualitatively suggests that significant perturbations of steady-state occur on shorter timescales ($10^5$–$10^6$ yr), like those created by Plio-Pleistocene glaciation. Our quantitative assessment of influx and outflux indicates that the Olympic Mountains could be in flux steady-state on long timescales ($10^7$ yr).

# 1 Introduction

The assumption of a balance between opposing processes has allowed geoscientists to use proxy measurements (like denudation rates) to constrain difficult to measure variables like rock uplift. This has given rise to the concept of steady-state landscapes or mountain ranges. Likewise, a steady-state (i.e., a mass balance) is commonly one of the boundary conditions in modelling studies investigating the evolution and dynamics of orogens in response to changes of other boundary conditions like climate or tectonic fluctuations (e.g., Batt et al., 2001; Stolar et al., 2007; Whipple and Meade, 2006; Willett, 1999). Two main types of steady-state are often used to interpret mountain building processes (e.g., Willett and Brandon, 2002): (1) Topographic steady-state, where the topography is invariant, because rock uplift and horizontal motion of material is balanced by denudation, and (2) flux steady-state, where the material influx (by accretion of sediment and rock) is balanced by the material outflux (by denudation) from a mountain range. The assumption of steadiness is both spatial- and timescale-dependent so that for a given timescale, steadiness might only be achieved on a large, orogen-wide spatial scale, due to the spatial averaging of single processes acting on a small scale (e.g., catchment-wide sediment discharge vs. orogen-wide sediment discharge). Furthermore, a possible perturbation of steady-state is sensitive to the timescale it takes for orogens to respond to variations in crustal deformation or a change in climate. If the timescales required for a change in the influx and outflux are significantly different from each other, a deviation from steady-state is likely.

Likewise, studies from different orogens worldwide suggest strong variations in denudation and exhumation on million-year timescales. These variations can be linked to changes in the tectonic conditions (e.g., Adams et al., 2015; Lease et al., 2016), internal dynamics of drainage basins (e.g., Willett et al., 2014; Yanites et al., 2013), changes in the magnitude of precipitation (e.g., Lease and Ehlers, 2013; Whipple, 2009), or the onset of glaciation (e.g., Berger et al., 2008; Bernard et al., 2016; Ehlers et al., 2006; Glotzbach et al., 2013; Gulick et al., 2015; Herman et al., 2013; Herman and Brandon, 2015; Lease et al., 2016; Thomson et al., 2010; Thomson et al., 2013; Valla et al., 2011; Yanites and Ehlers, 2012).

Based on thermo-kinematic modelling of thermochronometric cooling ages, the Olympic Mountains, USA, (Fig. 1a) have been proposed to be in flux steady-state since ca. 14 Ma (Batt et al., 2001; Brandon et al., 1998). The approach of these studies was to assume flux steady-state along a two-

dimensional profile across the Olympic Peninsula as a precondition in order to derive the kinematics of

the model from the balance between accretionary influx (governed by the thickness of accreted sediment and plate convergence rate) and denudational outflux (as set by exhumation rates). Because the cooling ages can successfully be modelled with the used kinematics, the mountain range is then interpreted to be in flux steady-state. However, possible temporal variations in parameters like sediment thickness, plate convergence rate or exhumation rates were not considered in these studies. Likewise, the impact of Plio-

Pleistocene glaciation on the flux steady-state hypothesis has not been considered yet, although the range was extensively incised by glaciers (Adams and Ehlers, 2017; Montgomery, 2002; Montgomery and Greenberg, 2000; Porter, 1964) and experienced significant changes in climate conditions over the past 3 Myr (Mutz et al., 2018). Numerical modelling studies investigated the mechanics of the wedge by either considering fluvial erosion (Stolar et al., 2007) or glacial erosion (Tomkin and Roe, 2007). A significant

response of the orogenic wedge to glaciation was suggested (Tomkin and Roe, 2007) and recent studies proposed that exhumation rates in the Olympic Mountains increased due to Plio-Pleistocene glacial erosion (Herman et al., 2013; Michel et al., 2018). Resulting high sedimentation rates during the Quaternary increased the sediment thickness on the oceanic plate and seem to have caused a change in the deformational style of the offshore part of the wedge (Adam et al., 2004).

In this study, we test the hypothesis of flux steady-state in the Olympic Mountains, considering variations in both the material influx and outflux. First, we test the temporal steadiness of exhumation rates from bedrock cooling histories with a 1D thermo-kinematic model, capitalizing on new samples which have been dated with three to four thermochronometers (apatite and zircon (U-Th)/He and fission-track data; AHe, ZHe, AFT, and ZFT, respectively). Second, instead of assuming flux steady-state as a

precondition, we attempt to estimate both the accretionary influx and denudational outflux independently from each other. We particularly consider possible temporal variations in parameters affecting both fluxes by using published data of the off- and onshore sediment records, and exhumation rates from thermochronometry. With our new thermochronometry data we reveal a previously undetected temporal variation in exhumation rates due to a change in the tectonics (a reduction in plate convergence rates that

resulted in a decrease in exhumation rate), as well as the previously reported increase in exhumation rates related to the Plio-Pleistocene glaciation (reflecting a change in climate). Similarly, both material influx

and outflux are temporally variable, especially during the Quaternary. A quantitative comparison between both fluxes suggests that the Olympic Mountains could be in flux steady-state over longer timescales (e.g., $10^7$ yr), if a three-dimensional geometry is considered.

## 2 Background

### 2.1 Geology and glacial history of the Olympic Mountains

At present, the Juan de Fuca Plate subducts obliquely with respect to the overriding North American Plate (Fig. 1a) at 34 mm/yr at the latitude of the Olympic Mountains (Doubrovine and Tarduno, 2008). The forearc high of the subduction zone comprises (from north to south) Vancouver Island, the Olympic Mountains and the Oregon Coast Range, and lies west of a forearc low (e.g., Georgia Lowlands, Puget Lowlands) and the active volcanic arc (Fig. 1a). Seismic imaging suggests a flatter subduction angle beneath southern Vancouver Island and the Olympic Mountains (Hayes et al., 2012; McCrory et al., 2012), compared to areas in the north and south (Fig. 1a). The modern configuration of the subduction zone was established by the latest Eocene (e.g., Brandon and Vance, 1992) after accretion of the Coast Range Terrane to the North American continent (Fig. 1c). This terrane represents a large oceanic plateau and extends from the southern tip of Vancouver Island to Oregon (Eddy et al., 2017; Phillips et al., 2017; Wells et al., 2014).

The accretionary wedge of the subduction zone is exposed onshore within the Olympic Mountains (Fig. 1a) and is composed of Eocene–Miocene flysch (Brandon et al., 1998; Tabor and Cady, 1978). This part of the mountain range is known as the Olympic Structural Complex (Brandon et al., 1998) and is separated from the surrounding Coast Range Terrane by the Hurricane Ridge thrust fault (HRF; Fig. 1c), a major discontinuity traceable in seismic surveys (e.g., Clowes et al., 1987; Calvert et al., 2011). Minor sedimentary rocks of Eocene age (Eddy et al., 2017; Tabor and Cady, 1978) are contained within the Coast Range Terrane besides the predominant ~50 Ma old marine and subaerial basaltic rocks (Eddy et al., 2017). Exhumation of the range commenced at 18 Ma and since 14 Ma, the orogen is supposed to be in flux steady-state (Batt et al., 2001; Brandon et al., 1998; Pazzaglia and Brandon, 2001).

Plio-Pleistocene glaciation has strongly influenced the present-day appearance of the Olympic Mountains (Fig. 1b). During its maximum extent at ~14 ka, the Cordilleran Ice Sheet advanced from the

Coast Mountains of British Columbia and covered Vancouver Island and large parts of todays' continental

shelf (Booth et al., 2003; Clague and James, 2002). The Puget and Juan de Fuca lobes of the Cordilleran Ice Sheet surrounded the Olympic Mountains in the east/southeast and in the north, respectively (Fig. 1b). Alpine glaciers incised deep valleys in the landscape, particularly on the western side of the range (Adams and Ehlers, 2017; Montgomery, 2002), where piedmont glaciers almost reached the Pacific Ocean (Thackray, 2001). Glacial erosion varied across the range, as the location of the Pleistocene equilibrium

line altitude increases from 1000 m in the west to 1800 m in the east (Porter, 1964), due to a strong precipitation gradient (> 6000 mm/yr in the west, < 1000 mm/yr in the east). Determining the exact onset of glaciation in the Olympics has proven difficult, but the oldest deposits of the Cordilleran Ice Sheet in the Puget Lowland are as old as 2 Ma and deeply weathered alpine till on the west side of the Olympics is interpreted to be of the same age (Easterbrook, 1986).

**2.2 Previous thermochronometry studies in the Olympic Mountains**

Within the Olympic Mountains, an extensive dataset of thermochronometric cooling ages from bedrock samples (Figs. 1b and 2) exists for AHe (Batt et al., 2001; Michel et al., 2018), AFT (Brandon et al., 1998), ZHe (Michel et al., 2018) and ZFT (Brandon and Vance, 1992; Stewart and Brandon, 2004). These thermochronometer systems record cooling through a temperature range of ~60–240°C (e.g.,

Brandon et al., 1998; Farley, 2002; Gallagher et al., 1998; Reiners et al., 2004), as they have effective closure temperatures of 70°C, ~120°C, ~180°C and ~240°C, respectively, for a cooling rate of ~10°C/Myr (Ehlers, 2005). The interpretation of thermochronometric cooling ages from sedimentary rocks (such as in the Olympic Mountains) is often complicated when the cooling signal from the sediment source region(s) has not been reset due to reheating during subduction and metamorphism. If a sedimentary rock

sample has not had sufficient exposure to temperatures above the closure temperature of a given thermochronometer, the sample might retain cooling ages that represent the source region's cooling history (referred to as unreset) or might be a mixture of provenance cooling histories and the reheating process (incompletely reset sample). Determining, whether a sample is completely, incompletely or un-reset can be difficult and usually depends on the statistics of cooling age populations, derived from the

dated mineral grains (e.g., Brandon et al., 1998). The reproducibility of single grain (U-Th)/He ages from

a sample provides an indication of whether a sample is reset or not. This is typically determined with n=4–7 grains. For the fission track method, a larger number of grains is typically dated (n=20–100) to reduce the uncertainty in the final cooling age calculation. For samples with a large population, statistical methods can be applied to decompose the chronometer date distribution into different populations, and to determine if some portion of the sample is reset (Brandon, 1992, 1996). In the case where a sample is incompletely reset, a significant young age peak is determined and interpreted as the sample cooling age (e.g., Brandon et al., 1998).

In the Olympics, the youngest published reset AHe ages (≤ 2.5 Ma) can be found in the western and central portions of the mountain range, and there are two unreset samples in the east (Fig. 2a). The pattern of AFT ages is more complicated (Fig. 2b), and most reset and incompletely reset samples are located in the central part of the mountain range, whereas unreset samples are restricted to areas outside the central (high topography) part of the range. The youngest reported AFT samples (2–4 Ma) are incompletely reset samples and fully reset samples have cooling ages between 7 Ma and 27 Ma. ZHe data show a well-developed trend of unreset cooling ages at the coast and reset 5–6 Ma ages in the headwaters of Hoh and Elwha rivers (cf., Figs. 1b and 2c). Reset ZFT samples (~13–14 Ma) are confined to a small area east of Mt. Olympus (Fig. 2d).

Based on thermo-kinematic modelling, Michel et al. (2018) attributed the observed AHe and ZHe age pattern to an ellipse-shaped exhumation pattern (with highest exhumation rates in the central, high-topography part of the mountain range, Fig. 2e), as predicted for a mountain range situated in an orogenic syntaxis setting (Bendick and Ehlers, 2014). Here, a bend in the subducted slab creates a mechanical stiffening, which in turn leads to rapid and focused exhumation at the surface (Bendick and Ehlers, 2014). High uplift rates in the central, high topography part of the mountain range are also corroborated by topographic analyses (Adams and Ehlers, 2017) and denudation rates based on cosmogenic nuclides (Adams and Ehlers, 2018). Furthermore, modelling of particularly young AHe ages (<2.5 Ma) suggests that exhumation rates increased significantly by 50–150 % due to Plio-Pleistocene glacial erosion (Michel et al., 2018).

## 2.3 Offshore sediment record

Data constraining the sediment thickness on the Juan de Fuca Plate before incorporation of sediment into the accretionary wedge are summarized in Figure 3. Three boreholes were drilled into the blanketing sediments of the Juan de Fuca Plate during deep-sea drilling projects (ODP 888, ODP 1027 and DSDP 174; Fig. 3 and Table 1), and provide estimates of the sediment thickness and age constraints. The sediment thickness at the deformation front of the subduction zone has been estimated from three seismic studies (Adam et al., 2004; Booth-Rea et al., 2008; Han et al., 2016).

Most of the sediment is contained within two deep-sea sediment fans with different sediment sources. Today, sediment sources for the Nitinat Fan (offshore Vancouver Island and the Olympic Mountains) include detritus from Vancouver Island, the Olympic Mountains, and material delivered by the Fraser river system (Fig. 3), which drains large parts of the Canadian Cordillera including the British Columbian Coast Mountains (Carpentier et al., 2014; Kiyokawa and Yokoyama, 2009). The Astoria Fan offshore the Oregon coast is mostly fed by the Columbia River and is sourced by a large area in the interior of the USA (Fig. 3).

The total sediment thickness varies between 2600–3500 m at the deformation front and decreases rapidly to 600 or 900 m approximately 100 km away from the deformation front. At the locations of ODP 1027 and DSDP 174, up to 50–70 % of the total sediment thickness are Quaternary deposits, and sedimentation rates more than doubled during the Quaternary (from 80–110 m/Myr to 250–270 m/Myr, Table 1). At the location of ODP 888 the drilled 570 m of core were deposited over the past 600 kyr, suggesting very high sedimentation rates of 950 m/Myr compared to 400 m/Myr for the total sediment thickness of 2600 m at the location of the core (Table 1). As determined from detailed, stratigraphic analysis of core ODP 888, sedimentation rates are also highly variable during the Quaternary. Rates during glacial periods can be as high as 1900 m/Myr compared to 700 m/Myr during interglacials (Knudson and Hendy, 2009). At sites ODP 888 and 1027, the source region of the sediments has been the Canadian Cordillera for the past 3.5 Myr, which has not been affected by glacial-interglacial cycles (Carpentier et al., 2014; Kiyokawa and Yokoyama, 2009). The provenance of the sediments at DSDP 174 is mostly the Proterozoic Belt Supergroup in the interior of the USA and differs significantly from present-day detritus of the Columbia River (Prytulak et al., 2006). Hence, Prytulak et al. (2006) suggest that

deposition of the upper 630 m of sediment at this site and the build-up of the Astoria Fan were governed by glacial outburst floods.

## 3 Methods

We use a multi-method approach to assess flux steady-state in the Olympic Mountains. This includes thermochronometric dating, thermo-kinematic modelling of cooling ages to obtain exhumation rates, and independent estimates of accretionary influx and denudational outflux. We calculate the influx based on constraints of the incoming sediment thickness and plate convergence rate, and the outflux based on spatial constraints of exhumation rates within the Olympic Mountains. The procedure for each method is outlined below.

### 3.1 Thermochronometric methods

Our strategy with thermochronometric dating was (1) to obtain samples, which are multi-dated with up to four thermochronometer systems (because these are particularly sensitive to reveal variations in exhumation rate) and (2) to collect samples within vertical profiles in order to obtain estimates of the exhumation rate at the site of the respective profile. Therefore, we dated several literature samples with additional thermochronometer systems (Table 2) and we also present 19 new bedrock samples from vertical profiles (Fig. 4, Table 2) and two additional bedrock samples (OP1528 and OP1556; Fig. 2, Table 2) collected at an elevation of ~400 m, enlarging the existing ~400 m equal-elevation data of Michel et al. (2018). All new samples are sandstones of varying grain size. A sample transect at Mt. Olympus extends from the bottom of the Hoh Valley to the apex of the Olympic Peninsula (Mt. Olympus, 2428 m), covering ~2 km of relief (Figs. 4a and b). The Mt. Anderson transect starts in the upper reaches of the Quinault Valley and terminates on the flank of Mt. Anderson covering a total elevation difference of ~1600 m (Figs. 4a and c). The Blue Mountain transect is located in the northern part of the Olympic Peninsula close to Blue Mountain, covering an elevation difference of ~1300 m (Figs. 4a and d). All collected samples were dated with the AHe and ZHe techniques, three of these were dated by AFT, and two were dated by the ZFT technique. Additionally, we dated 13 samples from Michel et al. (2018) by

AFT and five by ZFT thermochronometry. This process yielded seven samples with AHe, AFT, ZHe and ZFT cooling ages (Table 2).

Standard mineral separation techniques (sieving, magnetic and gravimetric separation) were used to obtain apatite and zircon separates from crushed rock samples. For AHe and ZHe dating mineral grains were hand-picked and dated in the thermochronometry lab of the University of Tübingen, following the dating protocol of Stübner et al. (2016). The Ft-correction for apatite (Farley, 2002) and zircon (Hourigan et al., 2005) is applied to the measured amount of helium. The (U-Th)/He age equation is solved using the approach of Meesters and Dunai (2005). From each sample, we dated 4–7 apatite grains or 3–6 zircon grains and the results of single-grain analyses can be found in Tables S1 and S2. Our approach for assessing whether a sample is reset or unreset and the procedure for exclusion of outliers is explained in the supplementary material (Section S1.1). For reset samples, we calculate the arithmetic mean age from the accepted single-grain ages, which is reported in Table 2 along with a one standard deviation (1SD) uncertainty.

Fission-track dating of apatite and zircon was performed using the external detector and the ζ-calibration techniques (Hurford, 1990). Details about the treatment of the apatite and zircon mounts in the Tübingen thermochronometry laboratory can be found in Falkowski et al. (2014) and Falkowski and Enkelmann (2016). Table 3 contains the AFT and ZFT sample ages, and explains the procedure for assessing whether a sample is reset or unreset. Data for single-grain ages from fission-track dating of apatite and zircon are reported in Tables S3 and S4.

## 3.2 Thermo-kinematic modelling: model setup and boundary conditions

To interpret cooling histories recorded by our thermochronometers as exhumation histories, we used a modified version of the thermo-kinematic model Pecube (Braun, 2003), which contains a built-in Monte Carlo approach to resolve temporal variations in exhumation histories (Adams et al., 2015; Thiede and Ehlers, 2013). The model allows exploring possible exhumation histories for a particular sample by varying exhumation rates through time at defined time steps. The accuracy of a particular exhumation rate history is estimated by comparing modelled with observed cooling ages. More age constraints, and hence thermochronometer systems, lead to better resolved, modelled exhumation histories. Therefore,

although we report 21 new thermochronometric ages, we only used the seven samples, which have age constraints from AHe, AFT, ZHe, and ZFT in our modelling efforts (OP1513, OP1517, OP1533, OP1539, OP1551, OP1573, OP1582; Table 2).

Thermo-physical parameters chosen for the modelling are typical values reported for the sandstones of the Olympic Mountains (Table 4). We performed a sensitivity analysis in order to find the most suitable time step for our simulations and the results of that analysis can be found in the supplementary material (Section S2). Based on the analysis, a time step interval of 1 Myr seems to be most appropriate to use, given the range of our thermochronometry ages and their respective uncertainties. During further
modelling, we initiated the models at 20 Ma and used the time step interval of 1 Myr with a maximum testable exhumation rate of 6 km/Myr. For each sample, we ran 20,000 simulations (each corresponding to a different exhumation history) and assessed the goodness of fit between observed and modelled data for the respective exhumation history, using a reduced $\chi^2$-test. Here, sample ages $\tau_o$ were compared with modelled ages $\tau_m$, using the uncertainty of the sample age $\sigma_o$ for the number (N) of thermochronometer
systems available for the respective sample:

$$\chi^2 = \left( \left( \frac{(\tau_o - \tau_m)^2}{\sigma_o^2} \right)_{AHe} + \left( \frac{(\tau_o - \tau_m)^2}{\sigma_o^2} \right)_{AFT} + \left( \frac{(\tau_o - \tau_m)^2}{\sigma_o^2} \right)_{ZHe} + \left( \frac{(\tau_o - \tau_m)^2}{\sigma_o^2} \right)_{ZFT} \right) \cdot \frac{1}{N} \quad (1)$$

If $\chi^2 \leq 2$, a specific model run was accepted as good. The number of accepted exhumation histories is shown in Figure 5 for each sample. From the range of acceptable exhumation rates at each time step (shown as blue shaded areas in Fig. 5), we calculated the mean exhumation rate together with 1 standard
deviation for each time step (red/dashed lines and grey areas in Fig. 5). Although the model provides output for the entire model duration of 20 Myr, a meaningful exhumation rate can only be obtained for the time interval between oldest thermochronometric age of a sample and today (shown in Fig. 5).

      For our purpose, we focus on exploring temporal variations in exhumation rates and therefore use a 1D model, where each sample is modelled independently from each other. In a 1D model, heat transport
and movement of particles is only considered in the vertical dimension within a column of rock, ignoring topography. This mode of modelling was selected because it allowed us to efficiently perform thousands of simulations quickly in order to cover a large range of possible exhumation rates. The high number of exhumation histories accurately predicts our observed cooling ages and allows for a robust statistical

assessment of the best-fitting exhumation history. Previous publications addressing exhumation histories
in other orogens have also highlighted that 1D models are often sufficient to explain most of the signal
recorded in thermochronometric systems (e.g., Adams et al., 2015; Thiede and Ehlers, 2013). In the
Olympic Mountains, Michel et al. (2018) argued that exhumation histories for the thermochronometer
systems considered here can be well explained by vertical velocity paths, too. Because the spatial
resolution of our seven considered samples is poor and they are all from the interior part of the mountain
range (Fig. 4), we cannot further resolve the exhumation rates outside this area, making a 3D model very
difficult to validate. Therefore, we limit our interpretations to the better-resolved exhumation histories
from the 1D model and focus on the primary temporal changes, rather than paleotopography, or specific
differences in the exhumation rates between samples.

Five of the seven considered samples are from the same elevation range (400–580 m), but two
samples are from higher elevations (1360 m and 1500 m, Fig. 5). Large differences in elevation between
the samples can impact the direct comparison between them (e.g., it can affect how changes in exhumation
rate are recorded from location to location). However, we are not able to correct for this circumstance (by
using an age elevation relationship), and therefore try to consider this complication when interpreting our
exhumation rate histories from the different samples.

## 3.3 Methods for estimating flux steady-state

To assess the flux steady-state hypothesis of the Olympic Mountains, we need independent
estimates of the material influx and outflux over time. For this, we focus on the time period since 14 Ma,
which corresponds to the proposed establishment of flux steady-state (Batt et al., 2001; Brandon et al.,
1998). Flux steady-state requires that the material influx into the wedge equates the amount of accreted
material, removed from the subducting slab. We assessed the amount of accreted sediment (material
influx) with two approaches. First, we calculated the amount of sediment incorporated into the
accretionary wedge at the deformation front (Fig. 6a) during the 14 Myr period. Second, we compared
this amount of "expected" accreted sediment with the observed amount of sediment residing in the
accretionary wedge along two cross sections. The material outflux from the mountain range is estimated

using results from thermo-kinematic modelling, by equating modelled exhumation with denudation, which can then be integrated spatially and over the 14 Myr period.

The previous flux steady-state analyses in the Olympic Mountains were performed in two dimensions along a profile crossing the Olympic Peninsula. However, exhumation rates within the Olympic Mountains are known to vary spatially (Brandon et al. 1998; Michel et al., 2018). This suggests
that the outflux is spatially variable, depending on the location within the mountain range. Hence, we performed our flux analysis in three dimensions and the resulting geometries are summarized in Figure 6. The influx is calculated along the length of the deformation front, and for the calculation of the outflux we considered almost the entire area of the Olympic Peninsula.

### 3.3.1 Calculating the accretionary influx

We used a similar approach as Batt et al. (2001) to calculate the accretionary influx, but used a three-dimensional geometry and additionally considered temporal variations of the used variables. Assuming all sediments resting on the subducting oceanic crust are incorporated into the accretionary wedge, the volume of accreted sediment ($V_{sed}$) can be approximated using the porosity of the sediment $\eta$, incoming sediment thickness d, length of the coast l, the duration of subduction t, and the subduction
velocity perpendicular to the present-day deformation front $u_{per}$:

$$V_{sed} = (1 - \eta) \cdot d \cdot l \cdot t \cdot u_{per} \quad (2)$$

A limitation to this approach is the assumption that all sediment resting on the down-going plate is accreted. There is geochemical evidence that, at early stages of subduction at the Cascadia Subduction Zone, sediment has been incorporated into the mantle and been involved in the magmatism of the
315 Cascades Arc (Leeman et al., 2005; Mullen et al., 2017). However, there are no estimates on the amount of sediment transported into the mantle at present, and most sediments seem to be accreted, either at the deformation front or underplated at depth (Calvert et al., 2011).

The variable with the greatest uncertainty in this calculation is the sediment thickness back in time that has now been subducted below the Olympic Mountains. As discussed above (Section 2.3), the
320 present-day sediment thickness of 2.5 km is the product of increased offshore sedimentation during the Quaternary and the pre-Quaternary sediment thickness is difficult to determine. Following the approach

described in the supplementary material (Section S3.1), we estimated a pre-Quaternary sediment thickness of 1.5 km. In total, we calculated three different sediment volumes based on different sediment thicknesses (Table 5). Assuming a thickness of 1.5 km and 2.5 km for the 14 Myr period yields a minimum and maximum value for the accreted sediment volume, respectively, representing a sediment volume unaffected by Quaternary sedimentation (1.5 km) and a volume for a likely too high sediment thickness, using the modern thickness (2.5 km). Alternatively, we considered an increase in sediment thickness from 1.5 km to 2.5 km at 2 Ma, which likely yields the geologically most meaningful volume.

The porosity of the sediment stack depends on the thickness and decreases with increasing overburden. According to Yuan et al. (1994), the porosity at depth z of the sediment stack can be approximated by

$$\eta = 0.6 \cdot e^{-z} \quad (3)$$

Using this equation, we calculated mean porosities of 31% and 22% for our sediment thicknesses of 1.5 km and 2.5 km, respectively.

Because the dip direction of the present-day deformation front is 72° ($\Phi_{def}$) and we only considered accretion perpendicular to the deformation front, we corrected the convergence rate (u) by using the convergence angle ($\Phi$) between the Juan de Fuca and the North American plates:

$$u_{per} = u \cdot \frac{\sin(\phi)}{\sin(\phi_{def})} \quad (4)$$

Both convergence rate and angle are variable over time and, therefore, we capitalized on the plate reconstruction model of Doubrovine and Tarduno (2008) to estimate these parameters over the past 14 Myr. Values shown in Figures 6b and 6c were calculated using the East-West Antarctica plate circuit model from Doubrovine and Tarduno (2008) for two different rotation models (Farallon M1 and M2 in the original publication). This yields a range of possible convergence rates and angles, providing an uncertainty on the calculated sediment volume. The temporal resolution is given by the number of magnetic isochrons used for the plate circuit reconstruction by Doubrovine and Tarduno (2008). From the temporal evolution of the corrected convergence rate (Fig. 6b), we calculated the sediment volume $V_{sed}$ accreted during the 14 Myr period using Equation 2 and the parameters discussed above. For the length l in Equation 2 we assumed a value of 131 km, which corresponds to the length of the coastline in

the area of the exhumation rate pattern (Fig. 6d). The calculated sediment volumes are reported in Table
5.

### 3.3.2 Sediment volumes along cross-sections

We estimated the actual volume of sediment currently residing in the accretionary wedge along
two cross-sections, which are approximately 50 km apart (Profile 1 and 2 in Fig. 7). The lower boundary
of the accretionary wedge is the top of the subducting oceanic plate, which is constrained from the Slab
1.0 model (Hayes et al., 2012; McCrory et al., 2012). The upper boundary is defined by the present-day
topography/bathymetry (from 10 m- and 500 m-resolution digital elevation models, respectively) and the
Hurricane Ridge Fault (HRF). At the surface, the location of the HRF is adopted from a geologic map
(Tabor and Cady, 1978) and below the surface we use information provided by a seismic study at depths
of 22 km and 34 km (Calvert et al., 2011). The uncertainty related to the position of the HRF (error bars
at HRF nodes in Fig. 7) was propagated to estimate an uncertainty for the calculated sediment volumes.
Further explanation of this approach is given in the supplementary material (Section S3.2). Because the
location of the HRF is not resolved at greater depths, we truncate the area considered for volume
calculation at 34-km depth. Finally, the calculated volume is corrected for the porosity of the sediment
stack. Davis and Hyndman (1989) use porosities of 4–10% for sediments contained within the
accretionary wedge offshore Vancouver Island. Hence, we use an average porosity of 6% in our
correction.

### 3.3.3 Calculating the denudational outflux

In the absence of extensional faults, denudation acts as the prime mechanism for exhumation in
the Olympic Mountains. Therefore, exhumation can be equated with denudation and the denudational
outflux from the range can be obtained from the spatial and temporal integration of exhumation rates.

The exhumation histories presented in this paper (Fig. 5) are well-suited to resolve temporal
variations in exhumation, and hence provide qualitative information about variations in the denudational
outflux. The low spatial density of the seven considered samples prohibits a quantitative assessment of

the denudational outflux. To overcome this problem, we reverted to the pattern and exhumation rates suggested by Michel et al. (2018), providing good spatial coverage of almost the entire Olympic Peninsula (Fig. 6d). The total amount of exhumation, which is used for calculating the outflux and corresponds to the temporal integration of the exhumation rates, is similar within uncertainty in both data sets. For example, the modelled exhumation rate is sufficient to explain the ZHe age of 10.2 Ma for sample OP1513

in both studies (Michel et al., 2018 and this study).

Our outflux calculations are based on the spatial integration of the entire exhumation rate pattern displayed in Figure 6d, which is then temporally integrated over the 14 Myr period. Additional to a constant exhumation scenario, we also considered an increase in exhumation rates, which is related to an increase in erosion due to Plio-Pleistocene glaciation of the Olympic Mountains (Michel et al., 2018). In

Table 5, we report the denuded volumes for the case of constant exhumation rates, and for the two possible increase scenarios suggested by Michel et al. (2018), equating a 50% increase in rates occurring at 3 Ma or a 150% increase in rates occurring at 2 Ma. In order to account for the porosity of the denuded rocks, we corrected the denuded volumes by a porosity of 6%, the same value we applied in the estimation of the volumes in the sedimentary cross sections

**4 Results**

**4.1 Thermochronometry**

Along the Mt. Olympus elevation transect (Fig. 4b), AHe ages (1.9–3.7 Ma) overlap with each other within sample error (except for the uppermost sample). ZHe ages (4.8–8.5 Ma) show a similar behaviour (with the exception of the lowermost sample; Fig. 4b). AFT ages for two samples are 5.1 Ma and 6.2 Ma,

and the obtained ZFT ages of this transect are all unreset. Within the Mt. Anderson transect (Fig. 4c), AHe ages (1.5–3.9 Ma) increase with elevation up to an elevation of 1400 m and decrease between 1400 and 2100 m. ZHe ages vary between 6.5–8.9 Ma and one sample at ~1400 m has an AFT age of 7.8 Ma. For the Blue Mountain transect (Fig. 4d), AHe ages (3.6–30.1 Ma, and one unreset sample) do not show a clear correlation with elevation, but, interestingly, the uppermost sample yields the youngest age. ZHe

ages of dated samples of this transect are all unreset.

Clear spatial patterns for the multi-dated thermochronometer samples are observable (cf., Fig. 2 and 4). AHe ages are reset (apart from one sample in the north-east of the mountain range) and decrease towards the centre of the mountain range, where very young ages (< 2.5 Ma) can be found. Seven fully reset AFT samples (5.0–7.8 Ma) are confined to the centre of the range (samples OP1513, OP1517, OP1533, OP1539, OP1551, OP1573, OP1582), overlapping with the area of reset ZHe samples. The remaining eight AFT samples are unreset (Table 3 and Fig. 4). Two samples at the north and east coast (OP1502 and OP1510) have the youngest age peaks at 26 Ma (comprising 29% of the dates) and 36 Ma (35%), respectively. Samples from the western part of the mountain range (OP1521, OP1522, OP1527, OP1528, OP1531) have younger age peaks of 5–16 Ma (comprising 20–76% of the dates). Furthermore, the youngest age peak of these samples decreases in age towards the area of fully reset AFT samples.

We also collected samples (OP1527 and OP1528) close to sample locations with the youngest AFT ages of Brandon et al. (1998), which were reported as incompletely reset samples (with youngest peak ages of 3.9 and 2.3 Ma). In the original publication, only a small number of grains were dated (n=31 and n=12). To improve the statistics of these two samples, we merge our single grain ages with those of Brandon et al. (1998) and obtain more robust age distributions (n=134 and n=80; Table 3). The youngest peak ages of the age populations for the two merged samples are 7.4 Ma and 4.7 Ma (2–4 Myr older than age populations reported by Brandon et al., 1998).

ZHe ages constrain an area of reset ages (4.8–10.2 Ma) in the central, high-topography portion of the mountain range (light grey-shaded area in Fig. 4a). Five of these samples have AFT (5.1–7.8 Ma) and ZHe (4.8–8.9 Ma) ages that overlap within sample errors, implying rapid cooling (and hence fast exhumation) through both systems' closure isotherms. AHe ages of these samples are younger (1.7–3.9 Ma) and do not overlap with AFT ages, indicating that exhumation rates decreased after cooling below the AFT closure isotherm.

Of the seven samples dated with the ZFT method, only sample OP1539 has a fully reset age (12.6 Ma). Together with data from Brandon and Vance (1992) and Stewart and Brandon (2004) this confines reset ZFT samples to a very small area east/southeast of Mt. Olympus, encompassing the headwaters of Elwha and Quinault rivers (area outlined with a red dashed line in Fig. 4a).

## 4.2 Exhumation histories from thermo-kinematic modelling

Between 13,000–17,800 simulations provide a good fit to the data for each of the seven samples
used in the thermo-kinematic modelling (Fig. 5). As expected, the four samples (OP1533, OP1539,
OP1551, OP1582; Fig. 5) with overlapping AFT and ZHe ages require fast exhumation rates of >3
km/Myr between 5 Ma and 8 Ma, followed by a reduction to <0.2 km/Myr at 5 Ma or 7 Ma. The reduction
of rates for sample OP1573 occurs at ~9 Ma. However, for this sample the AFT age has a larger
uncertainty, hence we consider the 5–7 Ma decrease in exhumation rates as a more robust signal. Six of
the seven samples (except for sample OP1517) also record an increase in exhumation rates at 2–3 Ma to
rates >1 km/Myr.

## 4.3 Estimating the flux steady-state balance

The calculated volumes of the accretionary influx depend strongly on the incoming sediment
thickness (Table 5). With our used three-dimensional geometry (Fig. 6a) volumes vary between ~70,000
km$^3$ (1.5 km), ~76,000 km$^3$ (increase from 1.5 km to 2.5 km at 2 Ma) and ~130,000 km$^3$ (2.5 km). The
estimated amount of sediment within the accretionary wedge varies depending on the position within the
wedge (Fig. 7). Offshore Vancouver Island, there is 950–1,000 km$^3$ of sediment within the wedge (Davis
and Hyndman, 1989), while on the Olympic Peninsula there is up to ~5,300 km$^3$ and 3,600 km$^3$ of
sediment within the central and southern parts of the mountain range, respectively. Our estimates of the
denudational outflux vary for the different exhumation rate scenarios (Table 5) and volumes range from
68,000 km$^3$ for constant exhumation rates to 75,000–82,000 km$^3$ for the exhumation scenario with
increasing rates.

## 5 Discussion

In the following, implications of the above described observations will be discussed in order to
assess the flux steady-state balance between accretionary influx and denudational outflux within the
Olympic Mountains. To do that, it is pivotal to have an understanding of both temporal and spatial
variations in exhumation of the Olympic Mountains. First, we elaborate on results from
thermochronometric dating, including the applicability of age-elevation relationships to reconstruct

exhumation rates in the Olympic Mountains (Section 5.1). Second, we analyse the general pattern of
exhumation based on the spatial distribution of cooling ages (Section 5.2). Third, we link
thermochronometric cooling ages with thermo-kinematic modelling, which reveals the temporal
evolution of exhumation rates (Section 5.3). Fourth, we discuss the outcome of our qualitative and
quantitative assessment of flux steady-state in the Olympic Mountains (Section 5.4). Finally, in Section
5.5, we elaborate on the limitations of the different approaches.

**5.1 Age-elevation relationships**

The cooling ages of samples collected from a quasi-vertical elevation profile (e.g., Fitzgerald et
al., 1993; Reiners et al., 2003) can be analysed by looking at the age-elevation relationship. Often, the
purpose is to determine an apparent exhumation rate by fitting a line through the data points when ages
are positively correlated with elevation. However, the prerequisite for this approach is that, over the lateral
extent of the sampled transect, no significant gradient in exhumation rates exists. This is not necessarily
given in the Olympic Mountains (Michel et al., 2018; see also Fig. 2e) and the new data represent this
complication (Figs. 4b–d).

At Mt. Olympus, the AHe and ZHe age-elevation relationships do show a positive correlation,
suggesting fast exhumation rates of ~1 km/Myr between ~8 and 2 Ma (Fig. 4b). The Mt. Anderson age-
elevation relationship for AHe shows a break in slope at ~1400 m and decreasing AHe ages at higher
elevations, and the large uncertainties of the ZHe ages limit an interpretation (Fig. 4c). While such an
'inverse' age-elevation relationship could be caused by a change in relief (Braun, 2002), we interpret it to
be a result of the strong spatial variation in exhumation rates along the horizontal distance of the transect
(e.g., rates increase from 0.25 km/Myr to 0.9 km/Myr over a horizontal distance of 15–20 km; Fig. 2e).
In the case of the Blue Mountain transect (Fig. 4d), we relate the non-correlation of AHe ages and
elevation to an incomplete resetting of the AHe system in this area. Here, some samples experienced high
enough temperatures to start, or even complete, resetting of the AHe thermochronometric system, causing
the observed variability in AHe ages. All ZHe ages from this transect are unreset, corroborating that this
part of the Olympic Mountains has not experienced high temperatures, compared to the other transects.
Indeed, the Blue Mountain transect belongs to the Coast Range Terrane (CRT), which is at a structurally

higher level compared to the accretionary wedge (Fig. 1c). In summary, the age-elevation plots support previous results of strong lateral variations in exhumation and incomplete resetting of thermochronometer systems in the outer part of the mountain range.

## 5.2 Pattern of exhumation

A well-constrained spatial pattern of exhumation is needed for calculating the denudational outflux. Looking at the spatial distribution of thermochronometric cooling ages provides qualitative information about the pattern of exhumation. In general, the distribution of thermochronometric ages indicates that in the Olympic Mountains the magnitude of exhumation increases from the coast to the centre. As discussed above, areas belonging to the Coast Range Terrane (close to the coast or the Blue Mountain area, where

unreset AHe ages can be found, Fig. 2a) correspond to the structurally highest parts within the range (Fig. 1c) and were not sufficiently reheated to reset the AHe system. Assuming a geothermal gradient typical for the Cascadia Subduction Zone of ~20 °C/km (Booth-Rea et al., 2008; Hyndman and Wang, 1993) and an AHe closure temperature of ~60–70°C, the cumulative exhumation magnitude since onset of exhumation at ~18 Ma cannot have been greater than 2–3 km.

The aerial exposure of the accretionary wedge (the Olympic Structural Complex, Fig. 1c) records exhumation from greater depths. Here, all samples yield reset AHe ages, requiring a minimum exhumation depth of 2–3 km. In the centre of the mountain range (encompassing the headwaters of Hoh, Queets, Quinault and Elwha rivers; Fig. 1b) the area of reset AFT ages approximately overlaps with the area of reset ZHe ages (Fig. 4a), requiring deeper exhumation, compared to the coastal part of the Olympic

Structural Complex.

The area east/south-east of Mt. Olympus (corresponding to the area of reset ZFT samples, Fig. 4a) has been exhumed from the greatest depths within the Olympic Mountains. For an average ZFT closure temperature of ~240 °C (Ehlers, 2005) and the above geothermal gradient this corresponds to a maximum exhumation from depths of 10–12 km, confirming previous estimates (Brandon and Calderwood, 1990;

Brandon and Vance, 1992).

In summary, the central, high topography part of the mountain range corresponds to the most deeply exhumed part. This corroborates the exhumation rate pattern (Fig. 2e) suggested by Michel et al. (2018),

the pattern of denudation rates based on cosmogenic nuclide dating (Adams and Ehlers, 2018), and results from topographic analysis (Adams and Ehlers, 2017), which all suggest that most of the exhumation/denudation occurs at this location. Hence, we use this pattern for the calculation of the denudational outflux.

## 5.3 Temporal variations in exhumation

Our new thermo-kinematic modelling revealed temporal variations in exhumation rates in the Olympic Mountains (Fig. 5). The decrease of exhumation rates at 5–7 Ma can be readily explained by the reduction in plate convergence rate and the change in convergence direction (Fig. 8). A Pacific-wide reorganization of plate movement at 5.9 Ma has been suggested (Wilson, 2002), and rapid uplift of the Oregon Coast Range at 6–7.5 Ma with a subsequent cessation in uplift has also been attributed to variations in the plate subduction parameters (McNeill et al., 2000). Furthermore, the volcanic record of the Cascadia Subduction Zone shows temporal variations, where the strongest volcanic activity lasted from 25 Ma until 18 Ma (du Bray and John, 2011). A period of volcanic quiescence, lasting from 17 Ma until 8 Ma, was then followed by increased activity, starting at ~7 Ma. A change in the stress field of the Cascadia Subduction Zone occurred at 7 Ma, which likely also affected the composition of the magmatism (Priest, 1990). Therefore, we interpret our observed 5–7 Ma drop in exhumation rates in the Olympic Mountains as a response to changes in the plate tectonic conditions.

In contrast, the increase in exhumation rates at ~2 Ma indicates a response to climatic rather than tectonic changes. As previously suggested by Michel et al. (2018), increased denudation due to the heavy glaciation of the mountain range led to an increase in exhumation rates by 50–150%, starting at 2–3 Ma. Our study corroborates these findings and shows that the observed young AHe ages require a recent increase in exhumation rates from slower rates (<0.2 km/Myr) lasting from ~7 Ma until ~2 Ma. Glaciation of the North American continent commenced at 2.7 Ma (Haug et al., 2005) and the oldest glacial deposits within the Olympics could be as old as 2 Ma (Easterbrook, 1986), overlapping with our modelled increase in rates at ~2 Ma. Due to the strong spatial variation of the Pleistocene equilibrium line altitude within the Olympic Mountains (Porter, 1964), glacial erosion likely also varied spatially, which could explain the different magnitude in increase of exhumation rates suggested for the different samples. Increased

offshore sedimentation related to glacially eroded sediment affected the deformational style of the offshore wedge leading to formation of west-ward dipping thrust faults, which changed at ~1.5 Ma (Adam et al., 2004; Flueh et al., 1998; Gutscher et al., 2001).

Taken together, these observations indicate that temporal variations in exhumation rates within the Olympic Mountains are subject to both changes in the tectonic and climatic conditions (as summarized in Fig. 8). The implication of these variations should be considered for the flux steady-state assessment.

## 5.4 Flux steady-state in the Olympic Mountains

## 5.4.1 A qualitative perspective

Several variables that affect both the accretionary influx and the denudational outflux show temporal variations. Exhumation rates decrease at 5–7 Ma and increase at ~2 Ma (Fig. 8) and since exhumation is primarily controlled by denudation, we equate these variations in exhumation with variations in the denudational outflux. According to the model of Doubrovine and Tarduno (2008), the plate subduction velocity decreased at ~6 Ma (see Fig. 6b) after an earlier major decrease at ~25 Ma, causing a decrease in the accretionary influx. Conversely, the accretionary influx increased significantly during the Quaternary due to high offshore sedimentation rates and increased sediment thicknesses as a result of effective glacial erosion on the North American continent (i.e., 50–70 % of the present-day sediment thickness on the subducting Juan de Fuca Plate consists of Quaternary-aged sediments, Table 1 and Fig. 3).

It follows that, qualitatively, both influx and outflux vary through time and are heavily influenced by the Plio-Pleistocene glaciation, which increased denudation rates and offshore sedimentation rates. However, we cannot quantitatively constrain whether variations in the influx and outflux on these short timescales (2–3 Myr) balance each other (and the system would still be in a flux steady-state). Interestingly, measured denudation rates based on cosmogenic nuclide dating (temporally integrating over the Holocene) suggest that modern denudation rates have not been significantly influenced by Plio-Pleistocene glaciation, but are mostly driven by tectonic rock uplift (Adams and Ehlers, 2018). The Holocene accretionary influx, however, is still affected by the increased sediment thickness since the

onset of glaciation. Hence, the current accretionary influx seems to exceed the denudational outflux in the Olympic Mountains.

### 5.4.2 A quantitative perspective

Here, we discuss the quantitative assessment of influx and outflux for the last 14 Myr (Table 5),
the time since when the Olympic Mountains are supposed to be in flux steady-state (Batt et al., 2001; Brandon et al., 1998). In our used geometry (Figs. 6a and d), we calculate the accretionary influx over a distance along the deformation front and the spatial exhumation rate pattern is integrated to infer the denudational outflux (Fig. 6d). Assuming an increase in sediment thickness at 2 Ma yields an accretionary volume (~76,000 km$^3$) similar to the denudational outflux (75,000–82,000 km$^3$). Assuming a maximum
sediment thickness of 2.5 km for the 14 Myr period yields an accretionary volume of ~130,000 km$^3$, which cannot be reconciled with our denudational outflux (Table 5). These results indicate that if temporal variations in the sediment thickness and denudation are considered, a reasonable balance between influx and outflux is attained. The previous flux steady-state analysis (Batt et al., 2001) was performed in two dimensions and used a constant sediment thickness of 2.0 km. We also performed an influx and outflux
calculation using a two-dimensional geometry to tie in with previous work (see appendix below). However, the results from the two-dimensional analysis suggest that for an area with spatially variable exhumation rates like the Olympic Mountains a three-dimensional geometry yields a more accurate prediction of influx and outflux.

Sediment volumes integrated along the cross-sections (Fig. 7) also provide an interesting
perspective on the accretionary influx in the Olympic Mountains. These volumes are not directly comparable with the influx/outflux volumes discussed above (calculated from 14–0 Ma), because the sediment contained within the cross-sections (Fig. 7) records accretion since the ~40 Ma onset of subduction (Brandon et al., 1998; du Bray and John, 2011). Furthermore, these estimates are minimum volumes, because the amount of material that has been eroded during the 40 Myr period is not considered.
Nevertheless, the amount of sediment currently residing in the accretionary wedge is variable along strike of the subduction zone (1000–5400 km$^3$) and is highest below the central part of the Olympic Mountains (Fig. 7). This requires that parameters affecting the accretionary influx (like plate subduction velocity or

sediment thickness) are highly variable over short distances (Profile 1 and Profile 2 are only 50 km apart; Fig. 7). Another explanation might be that considering accretion only perpendicular to the deformation is an oversimplification and another velocity component also contributes to material transport (see Section 5.5). This is in accordance with the conclusion drawn above that considering flux steady-state in a two-dimensional scenario (as it is done with the cross-sections) leads to ambiguous results.

In summary, the assessment of flux steady-state in the Olympic Mountains is non-trivial and several scenarios are possible. From a qualitative viewpoint, flux steady-state is probably not achieved on short timescales (few Myr), because the thickness of incoming sediment, plate subduction velocity, and exhumation rates show strong temporal variations on timescales of 2–3 Myr. From a quantitative viewpoint, influx and outflux volumes equate each other over longer timescales (i.e., 14 Myr), if influx and outflux are considered in three dimensions.

## 5.5 Restrictions and limitations of our approaches

In the sections above, we discussed exhumation in the Olympic Mountains and the results from our flux calculations. In the following section, we want to elaborate on possible restrictions or limitations in our approaches.

With our 1D modelling, we revealed strong temporal variations in exhumation rates (Fig. 5) related both to variations in tectonic and climatic conditions (Fig. 8). However, two of our modelled samples (OP1513 and OP1517) do not display the decrease in exhumation rates at ~5–7 Ma. These are from the Elwha valley (Fig. 4), in contrast to the five samples displaying the decrease, which are located in the western part of the mountain range. This suggests that the response of the orogenic wedge to a variation in the tectonic conditions affects only parts of the wedge and might be controlled by discrete structures. Further sampling and thermochronometric dating would be required to localize possible faults. Furthermore, this places a limitation on the application of a refined 3D model, because it requires to constrain parameters such as fault location or displacement on these faults. Besides the importance of single structures, the general pattern of deformation in the Olympic Mountains should still be viewed as controlled by the geometry of the subducted plate (Adams and Ehlers, 2017; Adams and Ehlers 2018; Brandon and Calderwood, 1990; Michel et al., 2018).

Regarding our flux analysis, we based our calculations on the volume of accreted sediment within a certain time (governed by the sediment thickness and the plate convergence rate) and the amount of denuded material (governed by the exhumation rates). As we mentioned in Section 3.3.1, a variable with great uncertainty is the sediment thickness over time, which has now been subducted below the Olympic Mountains. In the supplementary material (Section S3.1) we outlined our approach for assessing the pre-

Quaternary sediment thickness, which is used in our calculations. Although the reported 1.5 km sediment thickness seems to be a plausible value, we note that this value is afflicted with uncertainties and might have been higher. Nonetheless, our proposed balance between influx and outflux is still tenable, if the pre-Quaternary sediment thickness deviated from the assumed 1.5 km. I.e., we suggested an influx volume of 75–78 x $10^3$ km$^3$ and calculated outflux volumes between 75 x $10^3$ km$^3$ and 82 x $10^3$ km$^3$ (Table 5), so

even an additional influx volume due to a thicker, unnoticed sediment thickness could be balanced with our calculated outflux volumes. Another simplification in our calculation is the assumption of a spatially uniform sediment thickness over the considered length. Figure 3 shows that the sediment thickness along the deformation front is variable and is highest in the Nitinat and Astoria fans. However, an attempt to reconstruct along-strike variations in sediment thickness over time is challenging and would introduce

further uncertainties, and thus, we assume an average, constant thickness.

    During our influx calculations, we did not distinguish between different modes of accretion, such as frontal accretion or underplating. Batt et al. (2001) concluded that most accretion occurs at the front of the wedge. However, a recent seismic study showed that sedimentary underplating is taking place below the Olympic Mountains (Calvert et al., 2011). For our approach, the mechanism of accretion does not

matter, because we are only interested in whether mass is balanced over the entire wedge and not at a specific point. As indicated, this is a limitation of our approach and might lead to an overestimation of the actual influx volume, because we do not account for the amount of sediment transported towards the mantle.

    Flux steady-state implies that the outflux from and influx into a mountain range balance each

other. An inherent assumption is often that the material removed from a mountain range (the outflux) again enters the mountain range via the influx, which consists of the denuded material from the same source. So in case of an accretionary wedge, this implies that sediment is recycled and the system behaves

as a closed system. As we described in Section 2.3 of the manuscript, the sediment currently entering the accretionary wedge of the Cascadia Subduction Zone is a mixture of sediment from different source regions (e.g., Olympic Mountains, Vancouver Island, Canadian Cordillera and in case of the Astoria fan the interior USA, Fig. 3). With the increased detrital input from the Cordilleran Ice Sheet from outside the Olympic Mountains, this effect became particularly pronounced since the onset of Plio-Pleistocene glaciation. Hence, our influx/outflux calculations for the Olympic Mountains do not represent a closed system, where the influx into the Olympic Mountains is solely controlled by the outflux out of the system. However, our calculations indicate that on long timescales (i.e., over 14 Myr) flux steady-state is attained, which might seem surprising given that the sediment thickness is governed by contributions from different source regions. We suspect that processes during sediment deposition, like redistribution by turbidity currents and redeposition in more proximal parts of the Juan de Fuca Plate, play an important role in the final sediment budget. As a consequence, the amount of sediment denuded from the Olympic Peninsula in a given time period (the outflux) is dispersed as it enters the ocean, so that for the same time period only a fraction of the sediment thickness (governing the influx) is composed of material originating from the Olympic Peninsula.

Variations in the geometry or extent of the accretionary wedge were also not included in our flux analysis. Since onset of subduction at the Cascadia Subduction Zone with the present geometry at ~40 Ma, the wedge must have grown over time in order to attain its present shape. As soon as a balance between accretion and erosion is established, the shape of an orogenic wedge remains constant, controlled by its critical taper (e.g., Davis et al., 1983). However, Adam et al. (2004) showed that the Cascadia accretionary wedge responded to increased offshore sedimentation during the Quaternary by development of west-ward dipping thrust faults, shifting the deformation front further seawards thereby increasing the extent and volume of the wedge. An important parameter contributing to the shape of the accretionary wedge is the angle of subduction, which is flatter below the Olympic Mountains (compared to areas north or south) due to the bend in the subducted slab (Fig. 1a). A reason hypothesized for bending the subducting slab is extension in the Basin and Range Province, starting in the middle Miocene (Brandon and Calderwood, 1990). All these points indicate that parameters controlling the size and volume of the accretionary wedge are both spatially and temporally variable. However, we cannot account for all of

these circumstances in our flux calculations, because they are difficult to constrain quantitatively from available observations. Furthermore, because we based our flux calculations only on volumes of accreted or eroded material over the 14 Myr period, a comparison between these two volumes itself should not depend on a change in the shape or extent of the accretionary wedge.

As we pointed out in Section 5.4.2, flux steady-state is obtained by using a three-dimensional geometry. However, we only considered the deformation front-perpendicular velocity component for our influx calculations. The different sediment volumes contained in the reported cross-sections (Fig. 7) could indicate that on long timescales additional velocity components must be considered. We can only speculate that margin-parallel transport, which is a contentious topic at the Cascadia Subduction Zone

(e.g., Batt et al., 2001; McCrory, 1996; Wang, 1996), also contributes to the accretionary influx. Present-day GPS velocities corroborate this hypothesis, indicating northward movement of coastal areas south of the Olympic Mountains (e.g., McCaffrey et al., 2013; Wells and McCaffrey, 2013).

To summarize, several parameters like the location of faults within the orogenic wedge, the sediment source region, the temporal evolution of the wedge geometry or margin parallel transport are

difficult to constrain from current observations. Although we emphasized that not all of these parameters affect our flux analysis, further knowledge of these will refine the current understanding of steady-state in the Olympic Mountains.

## 6 Conclusion

Our new data set of multi-dated thermochronometer bedrock samples together with thermo-kinematic modelling suggests that several mechanisms contribute to the evolution of the Olympic Mountains. Modelling of the observed AHe, AFT, ZHe, and ZFT ages shows that variations in both tectonic and climatic conditions result in temporal variations of exhumation rates. We revealed a hitherto unnoticed response of exhumation to the tectonic signal (a reduction in plate convergence rate causing a

drop in exhumation rates), which can also be observed in other parts of the Cascadia Subduction Zone. Plio-Pleistocene glaciation of the Olympic Mountains led to increased denudation, resulting in increased exhumation rates.

Our approach of assessing flux steady-state in the Olympic Mountains by estimating the material influx and outflux independently from each other is promising, but yields ambiguous results. The observed temporal variations in exhumation rate require a variation in the denudational outflux. Likewise, the accretionary influx is also temporally variable, because the plate subduction velocity and incoming sediment thickness are variable through time. Qualitatively, this suggests that flux steady-state is perturbed on short timescales by variations in the tectonic or climatic conditions. Our quantitative calculations of the influx and outflux show flux steady-state may be achievable over long timescales (i.e., 14 Myr). Contrary to a previous flux steady-state analysis in the Olympic Mountains, our calculated influx and outflux volumes only balance each other, if a three-dimensional geometry is considered.

This study demonstrates the timescale ($10^5$–$10^6$ vs. $10^7$ Myr) and spatial dependence of a steady-state assessment in an orogenic wedge. Furthermore, the tremendous effect of the Plio-Pleistocene glaciation is demonstrated, which is capable of significantly perturbing the development of an orogenic wedge, where both the influx and outflux are affected. Because we obtain flux steady-state for a three-dimensional geometry (but only consider velocities parallel to the subduction direction), more work is needed to constrain the role of material transport parallel to the deformation front. Such studies will lead to a better understanding of the development of orogenic wedges situated in a complex tectonic setting like the Olympic Mountains.

**Appendix: Two-dimensional flux steady-state analysis**

In Section 3.3 we performed our flux analysis in three dimensions due to the spatially variable exhumation rates (Fig. 6d). In the following, we also calculate the influx and outflux using a two-dimensional geometry, so that our calculations can be compared to those from Batt et al. (2001). Here, the accretionary influx occurs at a single location at the deformation front, and the sediment volume ($V_{sed2D}$) is obtained by using a slightly modified version of Equation 2:

$$V_{sed2D} = (1 - \eta) \cdot d \cdot t \cdot u_{per} \quad (5)$$

Variables are porosity ($\eta$), incoming sediment thickness (d), time (t) and the deformation front perpendicular convergence rate ($u_{per}$). The further procedure is identical to the procedure outlined in

Section 3.3.1. Although the volumes obtained with this equation have a unit of km$^2$, no great uncertainty is introduced if the analysis is expanded over a width of 1 km, which then yields values of km$^3$ and "true" volumes. Calculated volumes are 520–540 km$^3$ (for a 1.5 km thick sediment stack), 980–1020 km$^3$ (for a 2.5 km thick sediment stack), and 580–600 km$^3$ (for the increase in sediment thickness from 1.5 km to 2.5 km at 2 Ma).

The outflux calculations are not based on the integration of the entire exhumation pattern in Figure 6d, but rates are only integrated along the white line in Figure 6. These integrals yield values of 68 km$^2$/Myr (constant rate), 103 km$^2$/Myr (50% increase in rate) and 171 km$^2$/Myr (150% increase in rate). After also integrating temporally (and assuming a width of 1 km, in order to get units of km$^3$, see above), the respective volumes of two-dimensional outflux are 900 km$^3$, 1000 km$^3$, and 1090 km$^3$.

A comparison of two-dimensional influx and outflux shows that the accretionary influx (~1000 km$^3$) only balances the denudational outflux (1060–1160 km$^3$), if an incoming sediment thickness of 2.5 km is assumed for the 14 Myr period. Hence, flux steady-state can only be obtained using a two-dimensional geometry, if an unrealistically high sediment thickness is assumed. Contrary to that, the three-dimensional geometry yields flux steady-state using a more reasonable sediment thickness (an increase in sediment thickness from 1.5 km to 2.5 km at 2 Ma). This indicates that assuming a two-dimensional geometry during the flux steady-state analysis is an oversimplification.

**Acknowledgements**

This work was funded by a European Research Council (ERC) Consolidator Grant (615703) to Todd Ehlers. During field work, we had invaluable help and assistance by Holger Sprengel, William Baccus, Jerry Freilich, Roger Hofmann, and the Olympic National Park rangers. We acknowledge Matthias Nettesheim for sharing the code used for evaluation of the tectonic plate reconstruction model, and the help of Willi Kappler during Pecube modelling. We thank Associate Editor David Lundbek Egholm for editorial handling of the manuscript. The comments by Phillipe Steer and one anonymous referee helped to improve and clarify this manuscript.

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

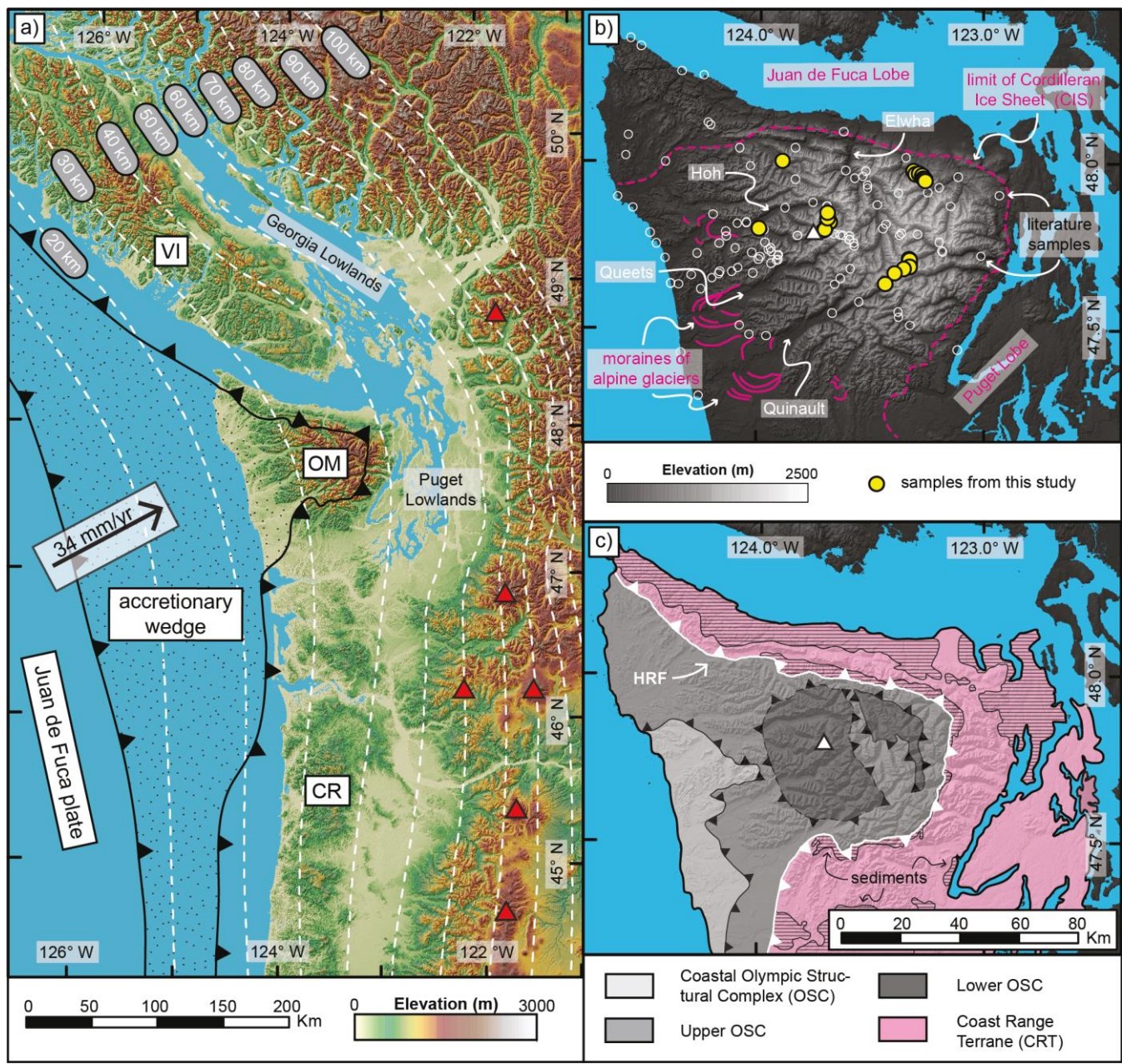

**Figure 1: a**) Overview map of the Cascadia Subduction Zone, showing the extent of the accretionary wedge. White dashed lines are contour lines for the top of the subducted oceanic plate from the Slab1.0 model (Hayes et al., 2012; McCrory et al., 2012), the black arrow indicates the present-day convergence rate and direction at the latitude of the Olympic Mountains (Doubrovine and Tarduno, 2008), red triangles denote the location of active volcanoes. VI = Vancouver Island, OM = Olympic Mountains, CR = Oregon Coast Range. b) Topography of the Olympic Mountains, major river valleys (Elwha, Hoh, Quinault, Queets) and major Quaternary features are indicated. Limit of the Cordilleran Ice Sheet from Porter (1964), alpine moraines after geologic map of Tabor and Cady (1978). Locations of samples from this study (filled yellow circles) and previous studies (open white circles) are indicated. The white triangle denotes the location of Mt. Olympus. c) Geologic and structural map of the Olympic Mountains after Tabor and Cady (1978) and Brandon et al. (1998). The line pattern indicates the occurrence of sediments within the Coast Range Terrane. HRF = Hurricane Ridge Fault.

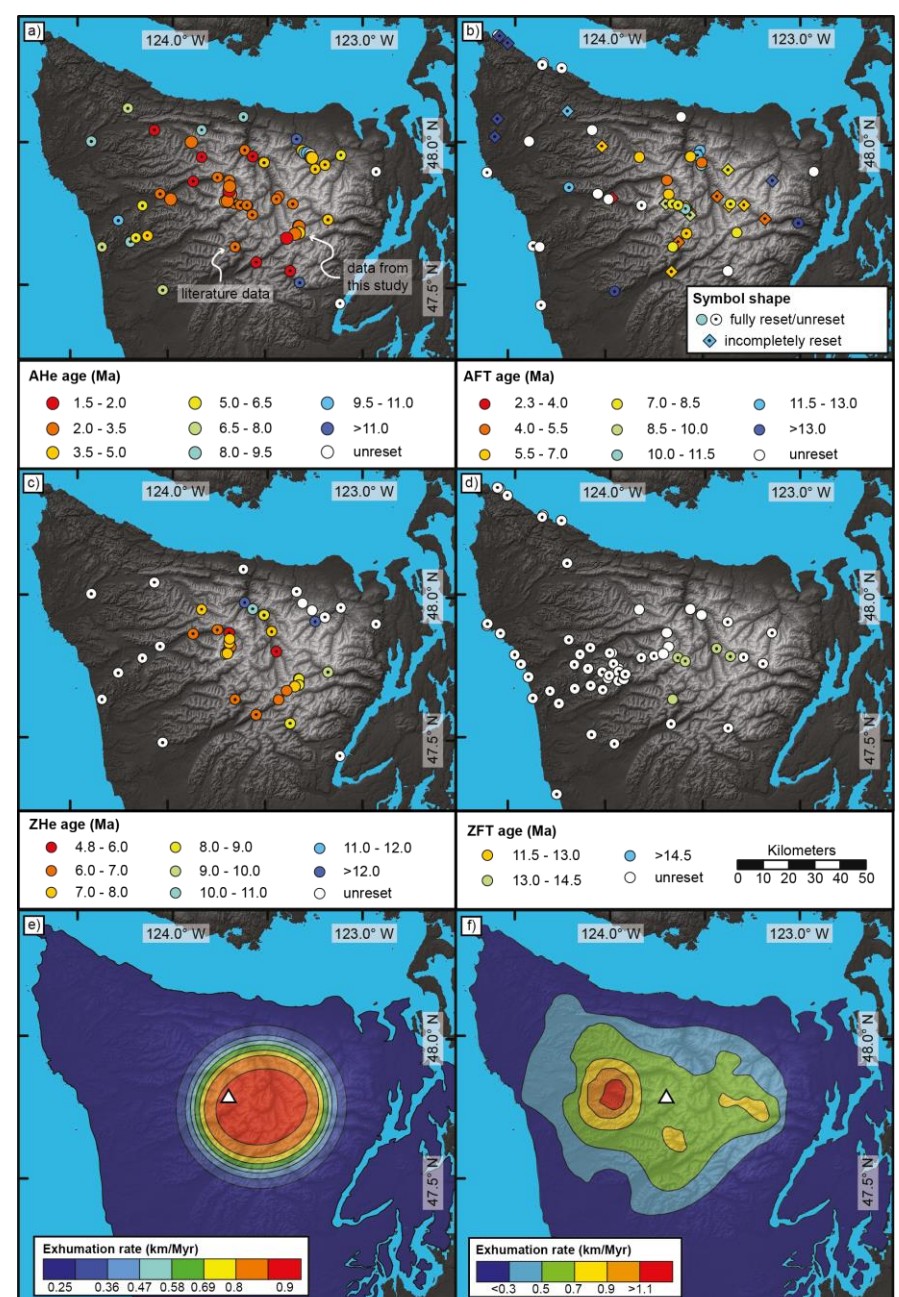

**Figure 2:** Map of new and previously published thermochronometric ages within the Olympic Mountains for a) AHe, b) AFT, c) ZHe and d) ZFT. Data from literature (Batt et al., 2001; Brandon et al., 1998; Brandon and Vance, 1992; Michel et al., 2018; Stewart and Brandon, 2004) are indicated by circles with black dot. Note that the colour coding of the symbols varies between panels. For AFT literature samples, the different reset states (fully reset, incompletely reset and unreset) are indicated by symbol shape. Maps of exhumation rates, as suggested by (e) Michel et al. (2018) and (f) Brandon et al. (1998). The white triangle denotes the location of Mt. Olympus.


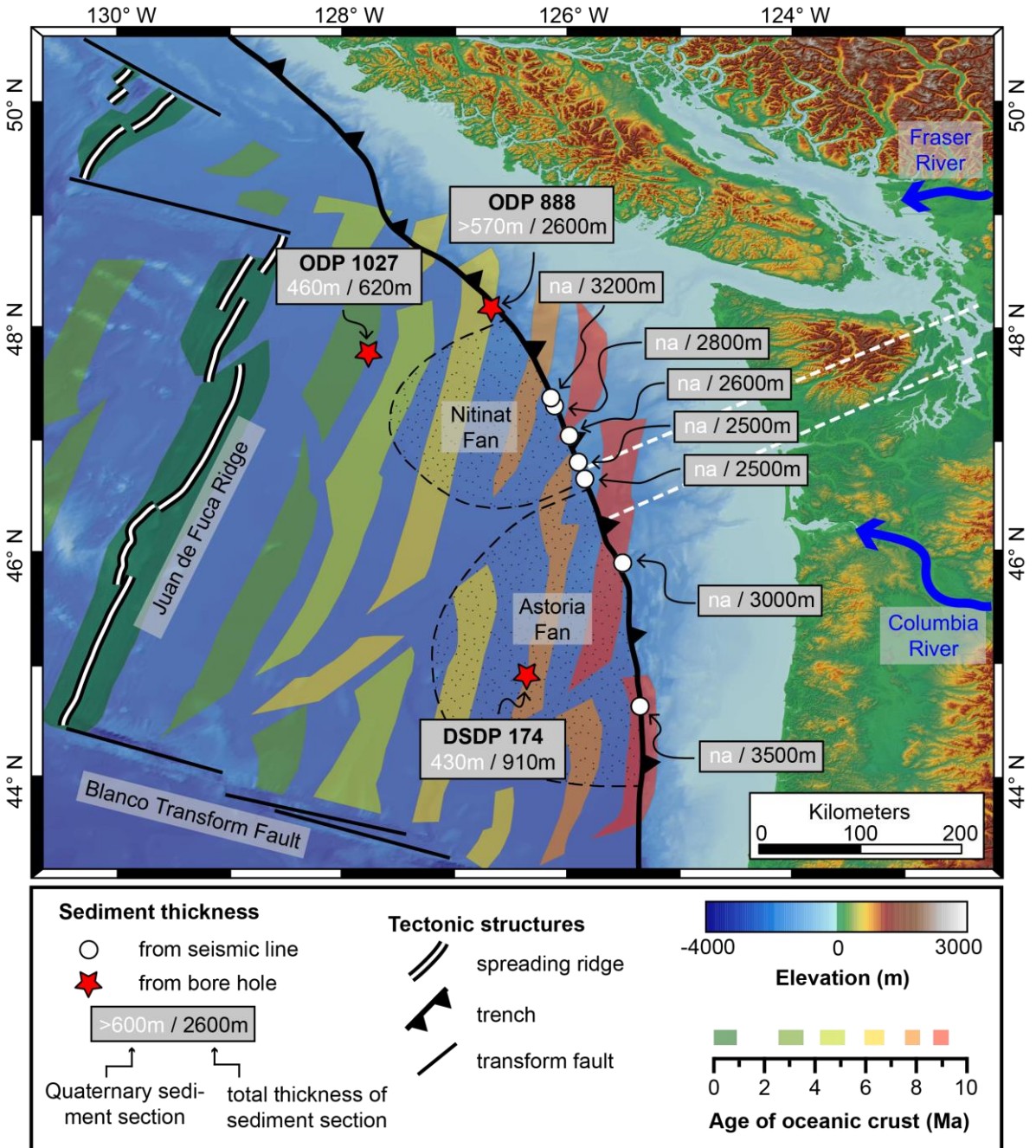

**Figure 3:** Map of the Cascadia Subduction Zone, showing the age of the oceanic crust (Wilson, 1993) and sediment thickness, estimated from sediment cores of the ocean drilling programs (holes ODP 888, OPD 1027 and DSDP 174) and seismic studies (Adam et al., 2004; Booth-Rea et al., 2008; Han et al., 2016). The amount of Quaternary sediment material estimated from cores is also included (Kulm et al., 1973; Su et al., 2000; Westbrook et al., 1994), more information about the drill cores is provided in Table 1. The locations of two major submarine fans (Nitinat Fan and Astoria Fan) are indicated by the dotted pattern. The Fraser and Columbia rivers are the main modern sediment sources for Nitinat and Astoria fans, respectively. White, dashed lines indicate the position of cross-sections presented in this study (cf., Fig. 7).

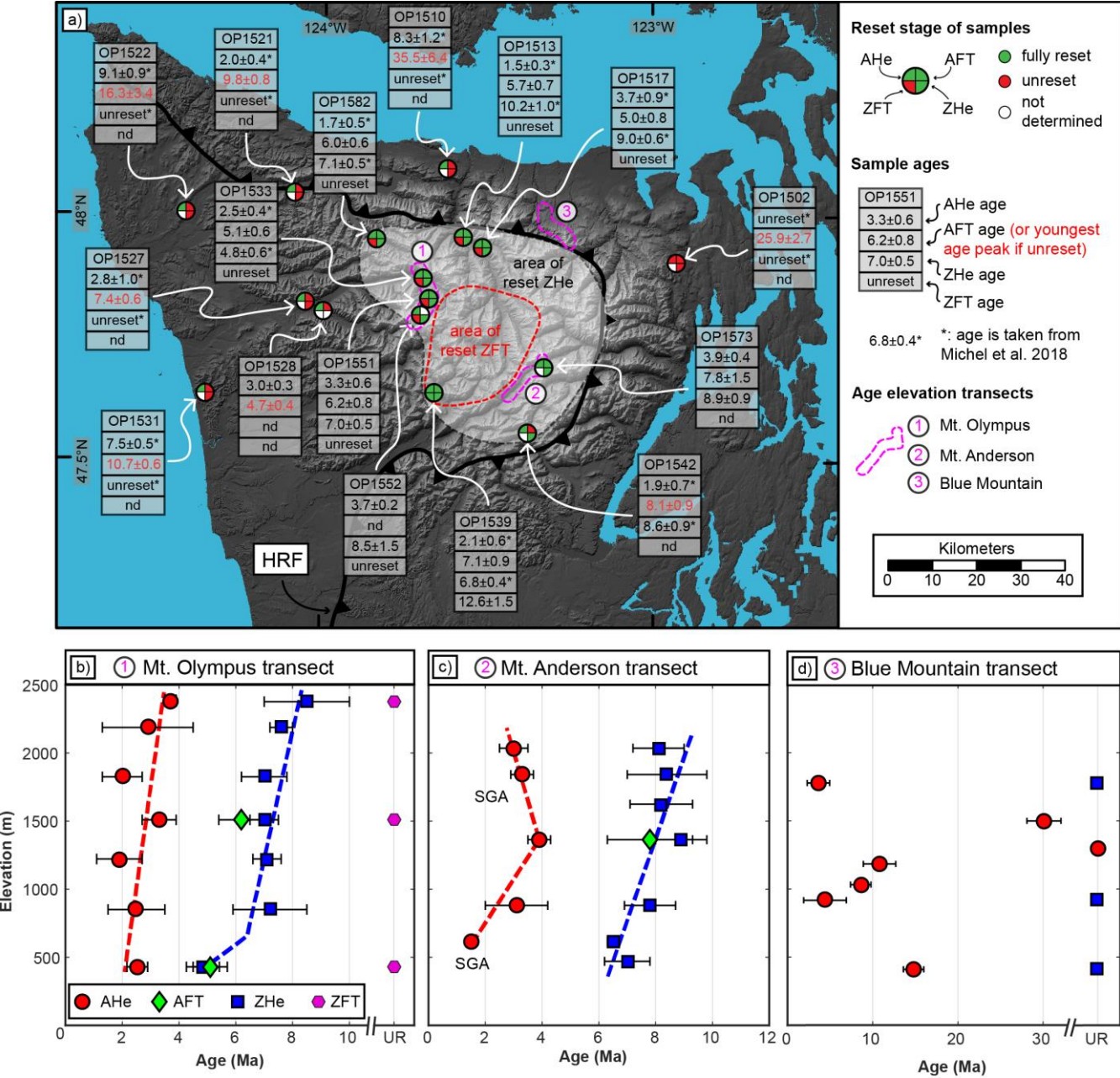

**Figure 4:** a) Map of samples, for which three to four different thermochronometer systems are available. The pie charts show the reset stage of a particular thermochronometer system for the sample. If AFT ages are unreset, the peak age of the youngest age population is given as sample age (see Table 3 for older populations). Ages denoted with an asterisk are taken from Michel et al. (2018). The Hurricane Ridge Fault (HRF) separates the rocks of the accretionary wedge from the surrounding Coast Range Terrane in the hanging wall. The locations of the three different elevation transects (Mt. Olympus, Mt. Anderson and Blue Mountain) are indicated on the map and the resulting age-elevation plots are shown in b) to d). In b) and c) the dashed coloured lines correspond to possible exhumation rates interpreted from the respective thermochronometer. All uncertainties are 1 standard deviation, SGA = single-grain age.


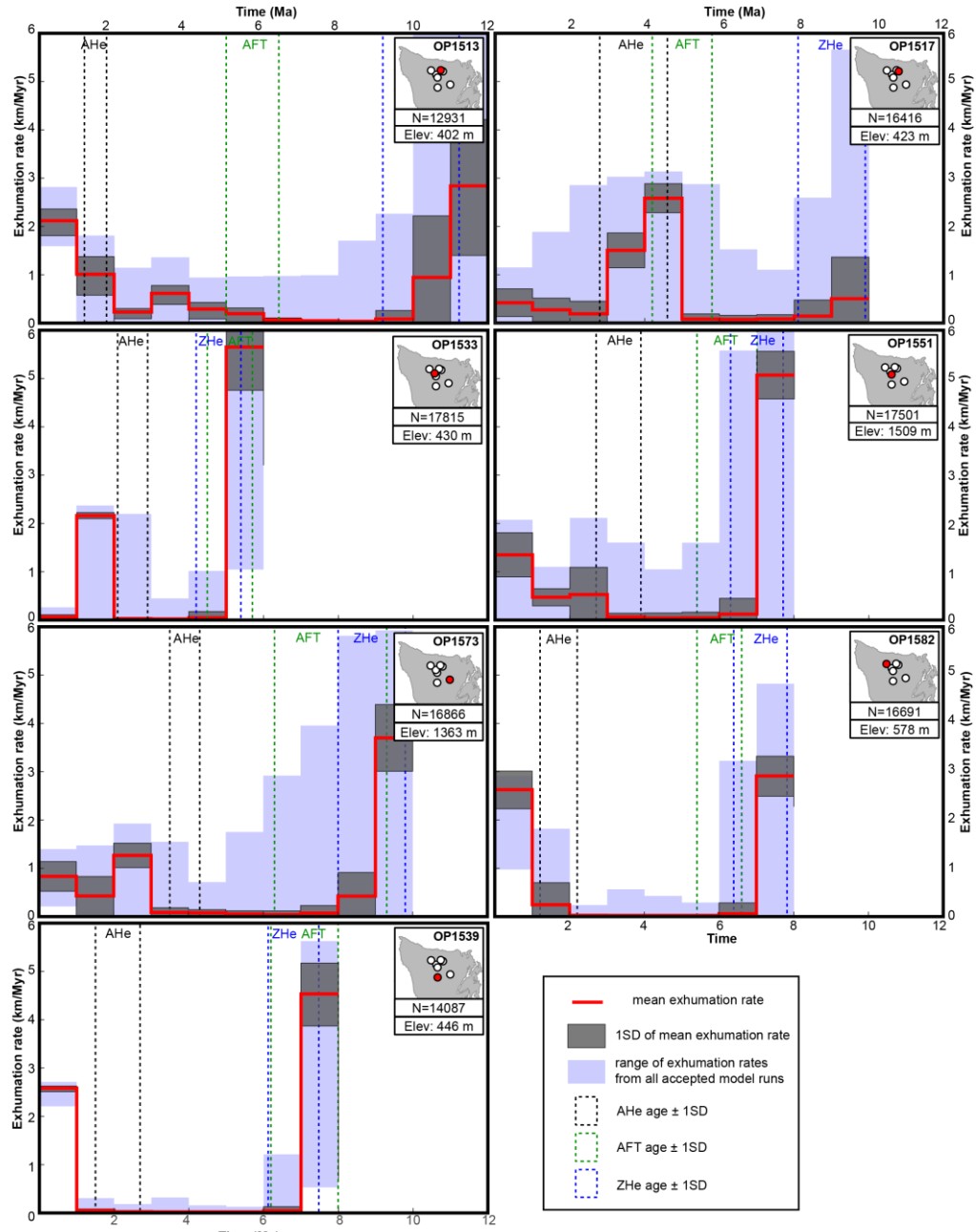

**1150**

**Figure 5:** Results from the thermo-kinematic Monte Carlo modelling for the seven considered samples (OP1513, OP1517, OP1533, OP1539, OP1551, OP1573, OP1582). Location of each sample within the Olympic Peninsula is shown, together with the respective elevation (Elev). The entire range of exhumation rates from the number of accepted model runs (N) is outlined by the blue shaded area, from which the mean rate and one standard deviation (1SD) is calculated at each time step. Black, green, and blue stippled boxes outline measured AHe, AFT, **1155** and ZHe ages of the samples with 1SD.

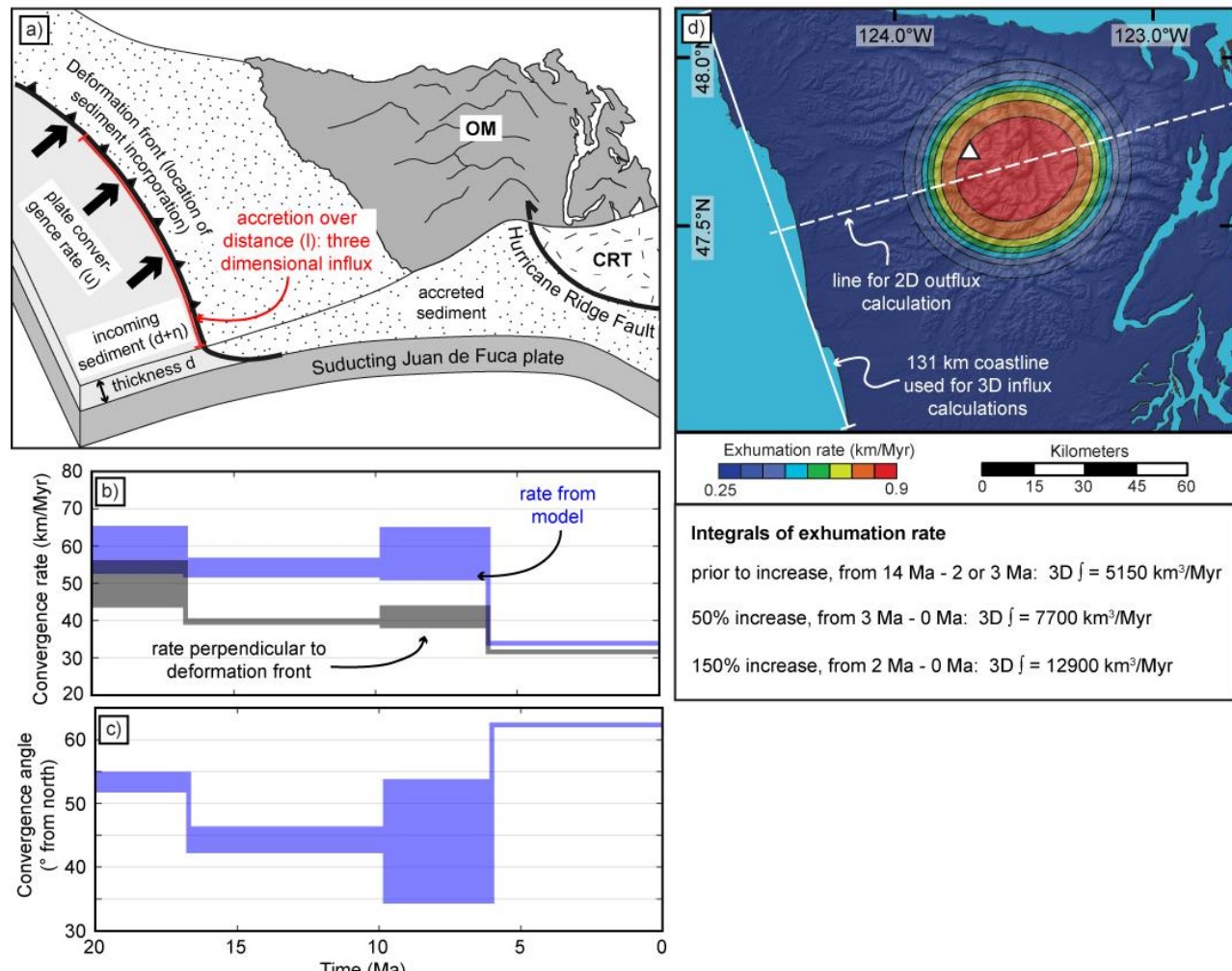

**Figure 6:** Constraints used for our quantitative accretionary influx and denudational outflux calculations. a) Cartoon illustrating our approach for calculating the accretionary influx. The influx corresponds to the sediment scraped off from the subducting Juan de Fuca Plate, and is governed by the plate convergence rate (u) and the incoming sediment properties (thickness d, porosity η). Because we use a three-dimensional geometry, accretion is considered along a length (l) within a vertical plane. This length corresponds to the length of the coastline indicated in panel (d). OM = Olympic Mountains, CRT = Coast Range Terrane. After Batt and Brandon (2002). b) Temporal evolution of the plate convergence rate used in the calculations, considering only the component perpendicular to the deformation front (black envelope), and the original output (blue envelope) from the plate reconstruction model of Doubrovine and Tarduno (2008). To provide an uncertainty for our calculations, we consider a range of convergence rates (comprising the width of the envelope) for each time step, based on two different rotation models in the model of Doubrovine and Tarduno (2008) (see text for details). c) Temporal evolution of the plate convergence angle (Doubrovine and Tarduno, 2008) used to correct the plate convergence rate in b). d) Exhumation rate pattern from Michel et al. (2018) used for our outflux calculations. The range of displayed rates (0.25–0.9 km/Ma) corresponds to the rates prior to the glacially induced increase in exhumation rates. The outflux is based on the spatial integration of the exhumation rate pattern. Values for the integrals are listed below the plot for the respective increase in exhumation and time. The white dashed line was used for integrating the exhumation rate using a two-dimensional geometry, which is further explained in the appendix.

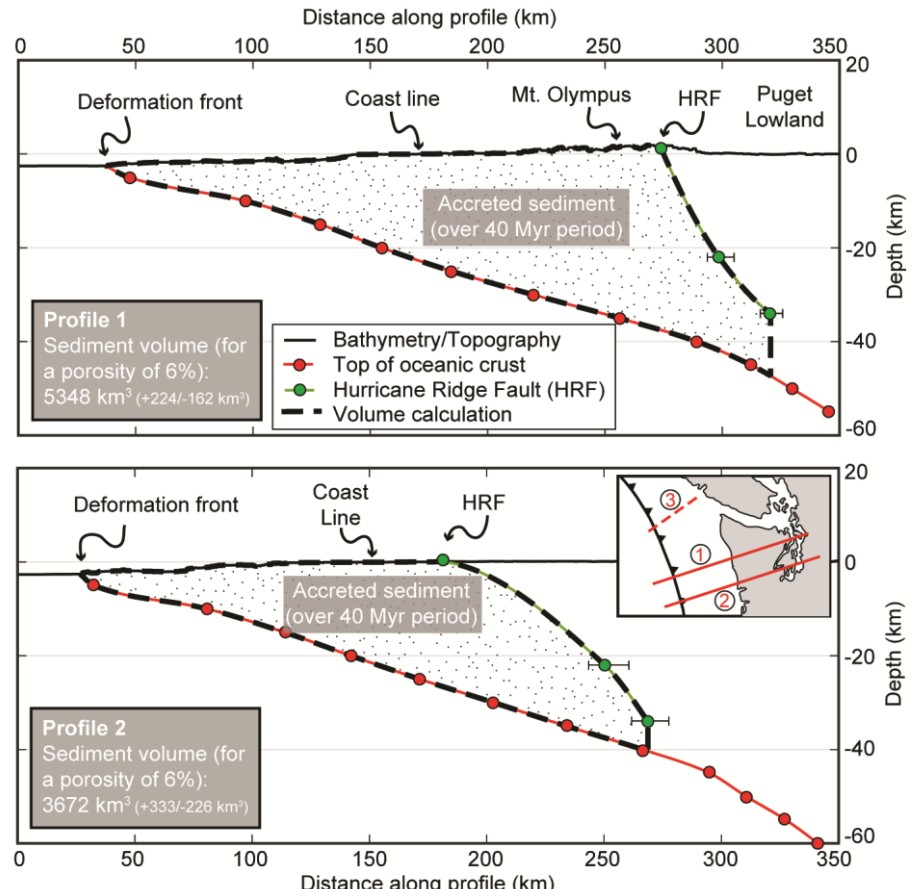

**Figure 7:** Sediment volumes calculated along two cross-sections spanning the Olympic Peninsula (Profile 1 and 2, vertical exaggeration=2, see inset for location). For explanation of the used procedure see text. The reported uncertainties for the volume are based on the uncertainties in the position of the Hurricane Ridge Fault (indicated with error bars at the respective symbol). Numbers in inset correspond to (1) = position of Profile 1, (2) = position of Profile 2, (3) = position of profile by Davis and Hyndman (1989), referred to in the text.

1175

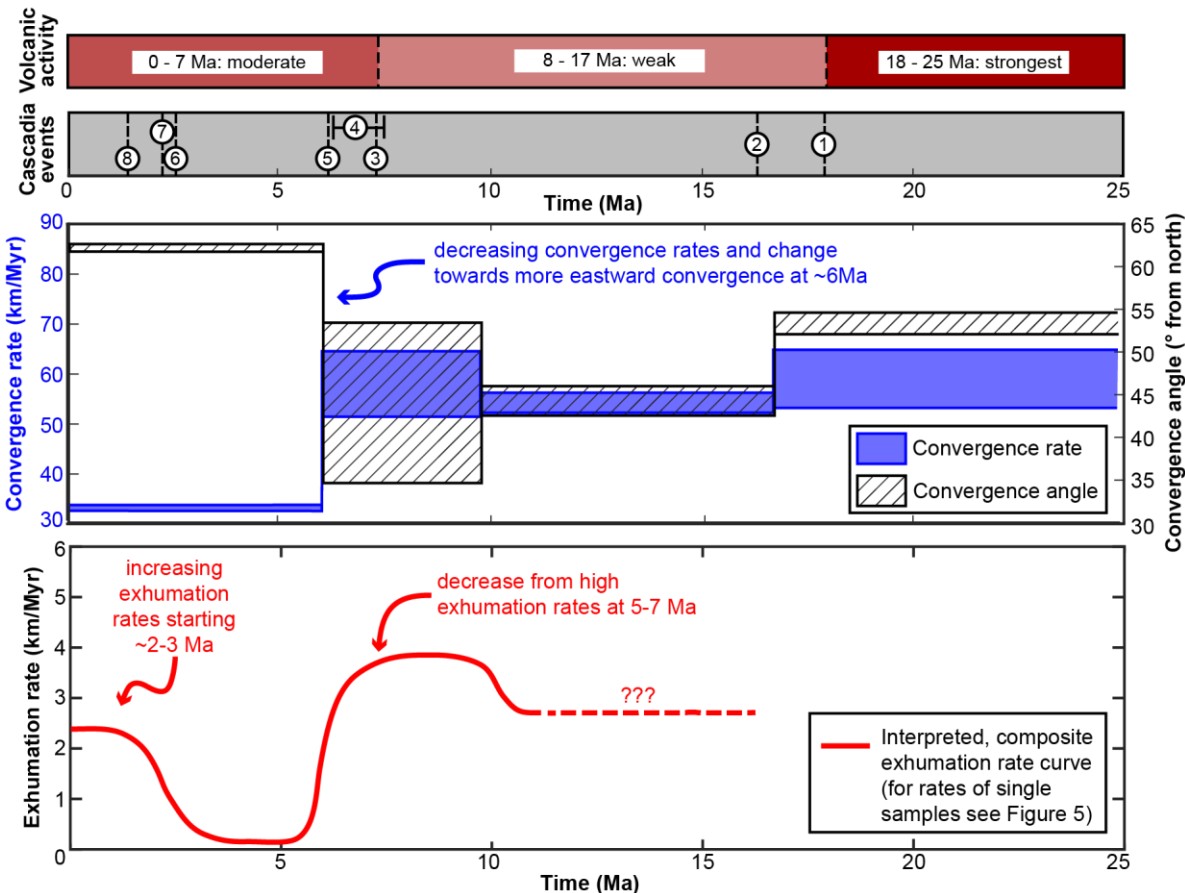

**Figure 8:** Summary of volcanic activity, tectonic and climatic events, and convergence rate and angle at the Cascadia Subduction Zone in comparison with our interpreted exhumation rates for the past 25 Myr. Exhumation rates are limited to the time interval covered by our thermochronometric ages (0–11 Ma). The curve depicts the interpreted evolution of exhumation rates, based on the modelling results shown in Figure 5 (see text for details). Volcanic activity after du Bray and John (2011). Tectonic and climatic events are (1) start of exhumation of the Olympic Mountains (Brandon et al., 1998), (2) onset of uplift of the Oregon Coast Range (McNeill et al., 2000), (3) rotation in stress field (Priest, 1990), (4) faster uplift in Oregon Coast Range (McNeill et al., 2000), (5) Pacific-wide plate reorganization (Wilson, 2002), (6) onset of North American glaciation (Haug et al., 2005), (7) onset of glaciation within the Olympic Mountains (Easterbrook, 1986), (8) change in the deformational style of the offshore accretionary wedge (Flueh et al., 1998). Convergence rate and angle from Doubrovine and Tarduno (2008).

**Table 1:** Data for the ocean drill cores shown in Fig. 3.

| Core | ODP 888 | OPD 1027 | DSDP 174 |
|---|---|---|---|
| Drilled/total sediment thickness[a] (m) | 570/2600 | 620/620 | 880/910 |
| Cored Quaternary sediment (m) | 570 | 460 | 430[b] |
| Maximum age of Quaternary sediments[c] (Ma) | 0.6 | 1.7 | 1.7 |
| Amount of Quaternary section of total core (%) | - | 74 | 47 |
| Age of oceanic crust[d] (Ma) | 6.5 | 3.2 | 7.5 |
| Quaternary sedimentation rate (m/Myr) | 950[e] | 270 | 250 |
| Pre-Quaternary/total sedimentation rate[f] (m/Myr) | -/400 | 110/190 | 80/120 |

**Notes:** For core ODP 888, information is taken from Westbrook et al. (1994), for ODP 1027 from Su et al. (2000), and for DSDP 174 from Kulm et al. (1973). Sedimentation rates are calculated in this study using the reported thicknesses and age constraints.
[a]: If total thickness exceeds drilled thickness, then the total thickness was estimated from seismic data (e.g., ODP 888).
[b]: Due to poor core recovery, the Plio-Pleistocene boundary can only be confined to be between 418 and 446 m.
[c]: Ages based on biostratigraphy. For cores ODP 1027 and DSDP 174, the Plio-Pleistocene boundary was recovered and an age of 1.7 Ma is used here as reported by Su et al. (2000).
[d]: For cores ODP 888 and DSDP 174, the age refers to the age of the oceanic crust and is taken from Figure 3 at the respective location of the core. For ODP 1027, the age refers to the age of the oldest sediment in the core taken from Su et al. (2000).
[e]: This rate is calculated for the recovered core interval, which only encompasses 600 ka.
[f]: Total sedimentation rate = total thickness divided by age of oceanic crust.

**Table 2:** Coordinates, elevations, and thermochronometric cooling ages for samples considered in this study.

| Sample | Latitude (°) | Longitude (°) | Elevation (m) | AHe ± 1SD (Ma) | AFT ± 1SD (Ma) | ZHe ± 1 SD (Ma) | ZFT ± 1SD (Ma) |
|---|---|---|---|---|---|---|---|
| **Mount Olympus transect samples** | | | | | | | |
| *OP1533[a]* | *47.87572* | *-123.69427* | *430* | *2.5 ± 0.4* | *5.1 ± 0.6* | *4.8 ± 0.6* | *unreset* |
| OP1550 | 47.81568 | -123.69601 | 1825 | 2.0 ± 0.7 | nd | 7.0 ± 0.8 | nd |
| *OP1551* | *47.82647* | *-123.68324* | *1509* | *3.3 ± 0.6* | *6.2 ± 0.8* | *7.0 ± 0.5* | *unreset* |
| OP1552 | 47.80155 | -123.71102 | 2377 | 3.7 ± 0.2 | nd | 8.5 ± 1.5 | unreset |
| OP1553 | 47.80377 | -123.70244 | 2188 | 2.9 ± 1.6 | nd | 7.6 ± 0.4 | nd |
| OP1554 | 47.83979 | -123.69330 | 1222 | 1.9 ± 0.8 | nd | 7.1 ± 0.5 | nd |
| OP1555 | 47.85457 | -123.69194 | 851 | 2.5 ± 1.0 | nd | 7.2 ± 1.3 | nd |
| **Blue Mountain transect samples** | | | | | | | |
| OP1548[a] | 48.02186 | -123.34295 | 410 | 14.8 ± 1.2 | nd | unreset | nd |
| OP1557 | 47.98098 | -123.31173 | 917 | 4.4 ± 2.5 | nd | nd | nd |
| OP1558 | 47.97233 | -123.30092 | 1032 | 8.6 ± 1.2 | nd | unreset | nd |
| OP1559 | 47.97287 | -123.28636 | 1184 | 10.8 ± 1.9 | nd | nd | nd |
| OP1560 | 47.96709 | -123.27110 | 1324 | unreset | nd | nd | nd |
| OP1561 | 47.95783 | -123.26785 | 1500 | 30.1 ± 2.0 | nd | nd | nd |
| OP1562 | 47.95696 | -123.26078 | 1778 | 3.6 ± 1.3 | nd | unreset | nd |
| **Mount Anderson transect samples** | | | | | | | |
| OP1570 | 47.70483 | -123.32813 | 1624 | nd | nd | 8.2 ± 1.1 | nd |
| OP1571 | 47.71657 | -123.32927 | 2035 | 3.0 ± 0.5 | nd | 8.1 ± 0.9 | nd |
| OP1572[b] | 47.71473 | -123.32815 | 1842 | 3.3 ± 0.4 | nd | 8.4 ± 1.4 | nd |
| *OP1573* | *47.69400* | *-123.32765* | *1363* | *3.9 ± 0.4* | *7.8 ± 1.5* | *8.9 ± 0.9* | *nd* |
| OP1574 | 47.68899 | -123.35093 | 881 | 3.1 ± 1.1 | nd | 7.8 ± 0.9 | nd |
| OP1576[b] | 47.67451 | -123.39235 | 614 | 1.5 ± 0.2 | nd | 6.5 ± 0.2 | nd |
| OP1577 | 47.64185 | -123.43398 | 470 | nd | nd | 7.0 ± 0.8 | nd |
| **Equal-elevation samples** | | | | | | | |
| OP1502[a] | 47.90796 | -122.92804 | 325 | unreset | unreset | unreset | nd |
| OP1510[a] | 48.09852 | -123.62231 | 273 | 8.3 ± 1.2 | unreset | unreset | nd |
| *OP1513[a]* | *47.96015* | *-123.57273* | *402* | *1.5 ± 0.3* | *5.7 ± 0.7* | *10.2 ± 1.0* | *unreset* |
| *OP1517[a]* | *47.93891* | *-123.51376* | *423* | *3.7 ± 0.9* | *5.0 ± 0.8* | *9.0 ± 0.6* | *unreset* |
| OP1521[a] | 48.04832 | -124.08702 | 390 | 2.0 ± 0.4 | unreset | unreset | nd |
| OP1522[a] | 48.00530 | -124.41620 | 367 | 9.1 ± 0.9 | unreset | unreset | nd |
| OP1527[a] | 47.82500 | -124.05184 | 280 | 2.8 ± 1.0 | unreset | unreset | nd |
| OP1528 | 47.80681 | -123.99661 | 140 | 3.0 ± 0.3 | unreset | nd | nd |
| OP1529[a] | 47.78265 | -124.14257 | 343 | 6.2 ± 1.1 | unreset | unreset | nd |
| OP1531[a] | 47.63659 | -124.34966 | 50 | 7.5 ± 0.5 | unreset | unreset | nd |
| *OP1539[a]* | *47.64151* | *-123.65870* | *446* | *2.1 ± 0.6* | *7.1 ± 0.9* | *6.8 ± 0.4* | *12.6 ± 1.5* |
| OP1542[a] | 47.56001 | -123.37533 | 450 | 1.9 ± 0.7 | unreset | 8.6 ± 0.9 | nd |
| OP1556 | 48.00848 | -123.89398 | 470 | 3.3 ± 0.9 | nd | nd | nd |
| *OP1582[a]* | *47.95595* | *-123.83732* | *578* | *1.7 ± 0.5* | *6.0 ± 0.6* | *7.1 ± 0.5* | *unreset* |

**Notes:** Samples in italics are used for 1D thermo-kinematic modelling. Results from single-grain analyses for AHe and ZHe are reported in Tables S1 and S2, respectively. Further details for AFT and ZFT dating can be found in Table 3, and single-grain analyses for apatite and zircon are reported in Tables S3 and S4, respectively. 1SD = one standard deviation, nd = not determined.

[a]: AHe and ZHe ages of the respective samples are from Michel et al. (2018).

[b]: Reported sample AHe ages are single-grain ages, because the yield of suitable apatite grains did not allow to date more grains.

1220

1225

**Table 3:** Results from fission-track dating.

**Reset samples**

| Sample +Mineral | Grain ages (Ma) | $\chi^2$ (%) | N | reset state | Sample age ± 1SD (Ma) |
|---|---|---|---|---|---|
| OP1513 ap | 0.9–17 | 47 | 24 | R | 5.7 ± 0.7 |
| OP1517 ap | 0–13 | 25 | 17 | R | 5.0 ± 0.8 |
| OP1533 ap | 2–15 | 11 | 20 | R | 5.1 ± 0.6 |
| OP1539 ap | 3–31 | 21 | 21 | R | 7.1 ± 0.9 |
| OP1539 zr | 8–18 | 19 | 21 | R | 12.6 ± 1.5 |
| OP1551 ap | 0–16 | 76 | 21 | R | 6.2 ± 0.8 |
| OP1573 ap | 5–17 | 11 | 6 | R | 7.8 ± 1.5 |
| OP1582 ap | 0–19 | 55 | 22 | R | 6.0 ± 0.6 |

**Notes:** For AFT and ZFT, 20 grains per sample were dated in a first step and it was checked whether the sample passes the $\chi^2$-test and can be considered as reset (i.e., > 5%; an indication for belonging to the same age population, e.g., Galbraith, 2005). If so, the pooled $\zeta$-age was considered as the sample age and reported here. If a sample failed the $\chi^2$-test (i.e., < 5%), the sample is considered unreset and, in the case of AFT, 100 grains were dated if enough grains are available. The detrital age distribution was then decomposed into detrital age populations using BINOMFIT (Brandon, 1992, 1996) and the peak ages of those populations (with asymmetric error range for each age peak, corresponding to the 68% confidence interval, CI) are reported. For the ZFT method, the information whether the sample is reset or unreset is sufficient for this study and no further grains were dated. Fraction equals the amount of grains contained within the respective age peak. N=number of counted grains. Contrary to Brandon et al. (1998) we did not consider multiply or partially reset AFT samples but treated them as unreset, because our thermo-kinematic model can only be applied to fully reset samples.
[a]: Results for this sample are obtained by merging grains from our sample OP1527 (n=103) and sample AR39 (n=31) from Brandon et al. (1998).
[b]: Results for this sample are obtained by merging grains from our sample OP1528 (n=68) and sample AR40 (n=12) from Brandon et al. (1998).

**Unreset samples** — Age peaks of the age populations

| Sample +Mineral | Grain ages (Ma) | $\chi^2$ (%) | N | reset state | Age (Ma) | 68% CI (Ma) | | Fraction (%) | Age (Ma) | 68% CI (Ma) | | Fraction (%) | Age (Ma) | 68% CI (Ma) | | Fraction (%) |
|---|---|---|---|---|---|---|---|---|---|---|---|---|---|---|---|---|
| OP1502 ap | 10–630 | 0 | 94 | UR | 25.9 | -2.5 | +2.7 | 29.4 | 84.7 | -8.1 | +9.0 | 48.3 | 243 | -54.6 | +70.0 | 22.2 |
| OP1510 ap | 18–191 | 0 | 80 | UR | 35.5 | -5.4 | +6.4 | 34.5 | 52.6 | -6.0 | +6.8 | 52.9 | 100.3 | -23.6 | +30.7 | 12.6 |
| OP1513 zr | 17–82 | 0 | 23 | UR | 30.9 | -3.5 | +4.0 | 70.9 | 52.6 | -7.5 | +8.8 | 29.1 | - | - | - | - |
| OP1517 zr | 27–57 | 0 | 25 | UR | 33.7 | -8.0 | +10.5 | 15.5 | 41.4 | -4.5 | +5.1 | 84.5 | - | - | - | - |
| OP1521 ap | 0.5–499 | 0 | 103 | UR | 9.8 | -0.8 | +0.8 | 60.5 | 35.1 | -4.0 | +4.5 | 30.6 | 261.9 | -50.4 | +62.2 | 8.9 |
| OP1522 ap | 6–237 | 0 | 20 | UR | 16.3 | -2.8 | +3.4 | 20.2 | 41.8 | -3.5 | +3.8 | 59.7 | 130.1 | -32.7 | +43.5 | 20 |
| OP1527[a] ap | 1–992 | 0 | 134 | UR | 7.4 | -0.5 | +0.6 | 67.7 | 24.0 | -2.0 | +2.2 | 28.0 | 209.3 | -65.4 | +94.5 | 3.5 |
| OP1528[b] ap | 0.4–237 | 0 | 80 | UR | 4.7 | -0.4 | +0.4 | 75.7 | 14.6 | -2.2 | +2.5 | 24.1 | - | - | - | - |
| OP1531 ap | 6–684 | 0 | 100 | UR | 10.7 | -0.5 | +0.6 | 50.6 | 30.2 | -1.8 | +1.9 | 40.1 | 149.0 | -21.7 | +25.3 | 6.5 |
| OP1533 zr | 29–106 | 0 | 23 | UR | 35.6 | -4.0 | +4.5 | 43.5 | 53.5 | -5.7 | +6.4 | 47.7 | 95.8 | -12.5 | +14.3 | 8.7 |
| OP1542 ap | 3–43 | 0 | 19 | UR | 8.1 | -0.8 | +0.9 | 77.1 | 27.9 | -3.6 | +4.2 | 22.9 | - | - | - | - |
| OP1551 zr | 9–57 | 0 | 23 | UR | 15.1 | -1.8 | +2.0 | 62.7 | 25.5 | -3.6 | +4.1 | 21.0 | 49.9 | -6.5 | +7.4 | 16.3 |
| OP1552 zr | 10–38 | 0 | 24 | UR | 14.0 | -1.5 | +1.7 | 79.7 | 32.3 | -3.7 | +4.2 | 20.3 | - | - | - | - |
| OP1582 zr | 28–68 | 0 | 23 | UR | 31.1 | -3.5 | +3.9 | 35.0 | 52.1 | -5.5 | +6.2 | 65.0 | - | - | - | - |

**Table 4:** List of parameters used for the Pecube modelling.

| Parameter | Value | Source |
|---|---|---|
| Thermal conductivity | 1.83 W m$^{-1}$K$^{-1}$ | average value for six drill cores in sediment material in the shelf offshore from Vancouver Island (Lewis et al., 1988) |
| Specific heat capacity | 1200 J kg$^{-1}$K$^{-1}$ | |
| Crustal density | 2700 kg m$^{-3}$ | |
| Mantle density | 3200 kg m$^{-3}$ | |
| Temperature at the base of the model | 400 °C | extrapolation to greater depths from temperature estimates based on heat flow measurements on the shelf (Hyndman et al., 1990; Hyndman and Wang, 1993; Booth-Rea et al., 2008) |
| Temperature at sea level | 8 °C | |
| Atmospheric lapse rate | 6.69 °C km$^{-1}$ | |
| Crustal heat production | 0.77 µW m$^{-3}$ | average value from drill cores on the shelf offshore Vancouver Island (Lewis and Bentkowski, 1988) |
| Model depth | 20 km | minimum thickness of the accretionary wedge below the Olympic Mountains (e.g., Davis and Hyndman, 1989) |

**Table 5:** Results from influx and outflux calculations using a three-dimensional geometry.

| | Accretionary influx over 14 Myr period [a] | | | Denudational outflux over 14 Myr period | | |
|---|---|---|---|---|---|---|
| | Minimum [a] (1.5 km) | Maximum [a] (2.5 km) | Increase at 2 Ma [a] (1.5 → 2.5 km) | Constant rates [b] | 50% increase at 3 Ma [b] | 150% increase at 2 Ma [b] |
| 3D | 68–71 x 10$^3$ km$^3$ | 128–133 x 10$^3$ km$^3$ | 75–78 x 10$^3$ km$^3$ | 68 x 10$^3$ km$^3$ | 75 x 10$^3$ km$^3$ | 82 x 10$^3$ km$^3$ |

**Notes:**

The entire procedure for calculating the influx and outflux is described in Section 3.3. The influx volumes are reported as ranges, because
10  minimum and maximum convergence rates (Figure 6b) have been obtained from the plate reconstruction model of Doubrovine and Tarduno (2008).

[a]: Sensitivity to incoming sediment thickness: The accretionary influx volume is calculated for three different sediment thicknesses, yielding a minimum volume (1.5 km thickness), maximum volume (2.5 km thickness), and a more realistic volume (where the volume increases from 1.5 km to 2.5 km at 2 Ma).

15  [b]: Sensitivity to an increase in exhumation rates: The denudational outflux volume is calculated assuming constant exhumation rates, and considering the increase in exhumation rates due to glacial erosion, with an increase by 50% at 3 Ma or an increase by 150% at 2 Ma. Exhumation rates are based on Michel et al. (2018) and displayed in Figs. 6d and e.