# Peer review of "How steady are steady-state mountain belts? - a re-examination of the Olympic Mountains (Washington State, USA)"

_Earth Surface Dynamics, 2018_

## Referee Comment (RC1) · Anonymous Referee #1 · 9 Oct 2018

The manuscript presents new thermochronological data from the Olympic Mountains in Washington State, USA. In addition, the paper presents a quantification of the influx and outflux of material to the mountain range over the last 14 million years, in order to discuss whether this accretionary wedge range is in a flux steady state. The influx of material into the accretionary wedge is based on knowledge from offshore sediment volumes and plate velocities, whereas the outflux is based on an exhumation map from previous thermochronological work in the range.

The topic is interesting and the overall finding represents a new scientific contribution. However, I find it peculiar how the authors present newly obtained data, apparently

without using them in the following calculation of the denudation pattern/outflux. As it is presented now, the paper appears quite fragmented, and one wonders what is really gained form including the new data. In this sense, the paper could just as well be split into two separate papers. One on the newly obtained thermochron data, and one on the flux in and out of the range.

Regarding the flux calculations, more justification is needed for the choice of sediment thickness. It seems like the 1.5 km is taken out of the blue. Also related to this, it should be better explained what the gain is from doing both the 2D and 3D approaches. Why not just do the 3D, and test this with the 2D cross sections? Isn't it obvious that a 2D approach is not ideal when a strong spatial gradient exists in exhumation perpendicular to the 2D section?

In addition, I have several comments outlined below:

Introduction: Lines 56-61: you don't need to summarize the conclusions here in the introduction. Methods: 150-153: Would be great to introduce here already what methods are used for the flux calculations. 182-: Although I appreciate that a 1D approach can give valuable results, I cannot stop to wonder why the authors did not take the full 3D approach, which is the core purpose of Pecube. To the best of my knowledge, all the authors are doing could be done in Pecube in 3D. Using all existing data, they could make an updated exhumation map for the region to be used in their calculations of the outflux. You should as a minimum discuss why it is not feasible to do a full 3D inversion in Pecube. 220-236: this section could be clearer and more up-front about the 2D vs. 3D approaches. Why not just use the 3D approach? What is gained from the 2D? this should be made clear. 252-255: the use of 1.5 km as the minimum for the previous thickness of offshore sediment is not properly justified. It is stated specifically that this is the largest unknown, and right now it seems you have grabbed this number out of thin air. Could there not have been more sediment earlier where you argue for a much higher exhumation rate? Lines 295-298: as mentioned above, it seems odd that you don't want to actually use the data you present here. Either you should use

all available data together, or you can just as well leave the new data out. Results: 312-318: I don't see AFT ages mentioned here? 345: should it not be OP1551 to be consistent with figures? 357: I would argue that the volumes vary with the location in the wedge geometry, latitude is irrelevant. Discussion: Lines 378-382. This is unclear. You start with: "In the absence of a strong lateral gradient"... and end with "due to the strong spatial gradient..."

Lines 401-402: references are needed here, or rephrase to avoid passive voice. Lines 480-482: more likely a lower outflux outside of the profile, is it not?

---

## Referee Comment (RC2) · Anonymous Referee #2 · 9 Oct 2018

This paper by Michel et al. aims at assessing the degree of steady-state of the accretionary prism of the Olympic Mountains. The authors used existing and new thermochronological data, to assess by an inverse model, the evolution of exhumation and denudation during the last 14 Myr that they equate to material outflux. For influx, they consider the rate of plate tectonic convergence multiplied by sediment thickness at the wedge front, and it is implicitly assumed that only frontal accretion contributes to material influx (a largely questionable hypothesis).

The question addressed by this paper is of interest for a large community, and I thank the authors for the efforts they have put in this manuscript. However:

1) The method used (inverse model based on Pecube with only 1D temperature and discretized at 1 Myr) is not optimal.

2) The influx reconstruction is not constrained or discussed well enough, in particular the contribution of fluxes other than frontal accretionary fluxes (such as isostasy, etc) or temporal changes in the structure of the wedge are not inferred.

3) The paper is sometimes confusing (e.g., some thermochronological data are not all used in the inversion, the 2D or 3D accretionary fluxes are not necessary (only in 3D or in 2D)).

4) The organization of the paper itself could be better to help its readability (for instance there is no results section on steady-state itself, which is surprising having read the title of the paper).

5) The addition of this paper, compared to established literature (Batt et al., 2001; Brandon et al., 1998; Michel et al., 2018) that already demonstrated a global steady-state over the last 14 Myr and change of exhumation rate at the onset of the glaciations, is not clear to me.

Detailed comments:

**L 32-35: A very minor comment: steady-state is time-scale dependent, but it is also spatial-scale-dependent. The likelihood of obtaining a steady-state, for a given time-scale, is likely decreasing when going to finer spatial-scales due to heterogeneities and variabilities in landscape dynamics or tectonics that might become more and more dominant in controlling averaged or integrated values (i.e., mean topography or mountain range sediment discharge). This applies to topographic and flux steady-states. This dual dependency of steady state to time and spatial scales (if correct), implies that defining a steady-state over a time-scale can be a correct or incorrect assumption, depending on the associated spatial-scale. So please be clear and explicit on which spatial-scale you investigate here.**

**L 51 - Please add a sentence to clearly state in the introduction, what are the main differences, in terms of methods and expected results, of this new manuscript compared to Michel et al. (2018) at Geology. Their abstract ends with: "However, the youngest AHe ages require a 50–150% increase in exhumation rates in the past 2–3 m.y. This increase in rates is contemporaneous with Pliocene-Pleistocene alpine glaciation of the orogen, indicating that tectonic rock uplift is perturbed by glacial erosion." Having read Michel et al. (2018), the differences are not clear to me at this stage of the paper (without having read the following sections).**

**L 202-207: I don't understand the need to use Pecube if neglecting topography and considering only a 1D model. An inverse modelling strategy using for instance QTQt (Gallagher et al., 2005 and so-on), that can jointly inverse samples from the same vertical profile, seems more appropriate here. This approach also has the benefit of not requiring any a priori time discretization. These advantages prevent the potential for both over interpretation and the introduction of artifacts in the inferred thermal histories.**

**L 230-236: What about isostasy or dynamic topography: are they not considered in the material influx? This need to be discussed.**

**L 249-251: There might be some circularity in the rational (to assess steady-state), as sediment thickness depends on the outflux and controls the influx.**

**L 295: "Our exhumation rates presented in this paper (Fig. 5) have a high temporal resolution". This statement might be overstated. No test was performed on the sensitivity of the inversion scheme and results to the time-step used (only 1 Myr was used). The data, especially individual samples, do not necessarily inform on a temporal evolution at a 1 Myr resolution. Is the inversion misfit better when changing temporal resolution? I would like to see some tests to determine the best time-step for the inversion (at least performed on one sample).**

**L 510 – section Results - Please add a sub-section at the end of the Results section, to present the results concerning flux steady-state analysis. This is the main ambition**

of this paper, and yet there is no result section on steady-state. This does not help the reader to get a clear message from reading this paper.

**L 367 – section Discussion - Please add a sub-section to present the limitations of the approaches used in this paper.**

**L 385-390: For the vertical profile with ZHe data, the change of polarity of the slope is not a robust feature. Models can be defined, satisfying all the ZHe ages and their uncertainties, without leading to a change of slope. However, for AHe, the change of slope seems robust.**

**L 451: It is assumed in this paper that the geometry of the accretionary wedge is constant and that other processes than tectonic accretion are negligible (an implicit assumption). Could you please discuss: 1) if there were some potential changes in the extent, volume and geometry of the Olympic Mountain accretionary wedge? 2) how you integrate isostasy or dynamic topography in your comparison of in- and out-fluxes? The isostatic response to erosion can generate uplift (with no associated influx in the presented model) and induce additional erosion. This need to be discussed in this manuscript (not as a perspective L533-536).**

**L 505: "In summary, the assessment of flux steady state in the Olympic Mountains is non-trivial and many scenarios are possible." The used datasets (thermochronological data, sediment deposits, geometrical structure, etc) are not sufficiently well resolved to offer a robust assessment of temporal changes in fluxes or in steady-state. Therefore, one could question the real addition of this paper compared to Batt et al. (2001), Brandon et al. (1998) or Michel et al. (2018) that have already demonstrated 1) a global steady-state over the last 14 Myr and 2) a potential change in exhumation rate with the onset of Plio-Pleistocene glaciations.**

Minor edits:

**L 14 : "We present 61 new thermochronometric ages" - Please add: mainly obtained**

from 21 new samples (or the correct number).

**L 37: “tectonic parameters” – please change by “tectonic conditions” (a parameter implies a quantitative framework/model that has not been defined yet).**

**L 39: “Plio-Pleistocene glaciation” - There is probably no need to limit the scope to the onset of Plio-Pleistocene glaciation, as older glaciations (for instance at the Eocene-Oligocene transition; Bernard et al., 2016; Thomson et al., 2013) might have also led to variations in denudation and exhumation.**

**L97: ref “Ehlers et al. 2005” : The closure temperature of these thermochonometers has been constrained in older papers than Ehlers, 2005. For instance: Gallagher et al., 1998; Farley, 2002; ...**

**L 162: “three/two” - What does three/two mean here?**

---

## Author Response (AR1)

**Authors' response to reviews for Manuscript esurf-2018-65**

"How steady are steady-state mountain belts? - a re-examination of the Olympic Mountains (Washington State, USA)"

By: Michel et al.

Dear Editor,

We acknowledge the time and effort the associate handling editor as well as the two reviewers have put into handling and reviewing our manuscript. The manuscript benefited significantly from the suggestions by the reviewers and we hope we sufficiently addressed the concerns of
the reviewers. Based on their suggestions, we made changes to the entire manuscript, and particularly rewrote and reformulated the introduction, the methods section as well as the discussion, where we also included a new section (Section 5.5). Furthermore, we conducted additional model simulations and provide the results from these in the electronic supplement.

In the following, we identify the five main concerns raised by the reviewers and we briefly
describe, how we addressed these. After this the point-by-point response to both reviewers can be found. This response includes our two comments already posted online during the discussion period of the paper, but now also indicates the discrete changes in the revised manuscript. At the end of this letter we provide the revised manuscript in a track-changes fashion, so that the editor and reviewers can readily see, what we changed in this resubmitted
version of the manuscript.

The main issues raised by the reviewers partly overlapped. We summarize these below, together with a brief response (a more thorough response is provided in the point-by-point section):

1) Inappropriate modelling strategy for our thermo-kinematic model
Reviewer#1 advocates that a full 3D model using Pecube would be more appropriate and reviewer#2 suggests to use thermal models like QtQt for our purpose. As we further elaborate in the detailed response below, our "simple" 1D model approach using Pecube with the Monte-Carlo algorithm is most suited for the main purpose: we want to demonstrate that exhumation is temporally variable. Setting up a new full 3D model
also considering the observed temporal variations requires to constrain many more parameters (which is partly not possible with the data available at present) and would be a paper on its own.
        In the revised manuscript we rewrote the methods section for the kinematic modeling (Section 3.2) and tried to better explain our purpose in using the 1D model. We also
conducted additional model simulations in order to investigate the timestep bias mentioned by reviewer #2, which we report in the electronic supplement of the manuscript.

2) 2D and 3D flux steady-state analysis

Both reviewers suggest to conduct the flux steady-state analysis only either in 2D or 3D. The previous analysis by Batt et al. (2001) was performed in 2D, hence we also provide a 2D analysis, to make results from both studies better comparable. However, as we show, with our approach flux steady-state is only attained using a 3D geometry. This is a new and unexpected observation, which has not been reported for the Olympic Mountains so far.

So, we still kept both the 2D and 3D analysis in the revised paper. However, we try to more clearly convey our purpose for doing so in the revised manuscript. We now explain in detail the approach of Batt et al (2001) in the introduction and how our approach is different. We also emphasize in the introduction and conclusion that the Olympic Mountains are only in steady-state, if a three-dimensional geometry is assumed.

3) Limitations of our approaches

This was particularly a concern raised by reviewer#2. We include a new chapter in the discussion (section 5.5) where we try to address the limitations inherent to our approaches, including thermo-kinematic modeling and parameters affecting our flux steady-state analysis (like sediment thickness and wedge geometry).

4) Pre-Quaternary sediment thickness for influx calculations

Reviewer#1 raised the concern that our value of 1.5 km for the pre-Quaternary sediment thickness used in our influx calculations is not well constrained.
We provide a detailed approach for estimating this difficult to constrain parameter in the electronic supplement of the manuscript (Section S3.2).

5) Novelty of presented results

Reviewer#2 raises the issue whether the outcomes from this work are new compared to the published work of Michel et al. (2018) and Batt et al. (2001).
To overcome this concern, we reformulated the introduction, which now includes a more detailed description of the previous approach of Batt et al. (2001). We now also highlight, which of our results and observations are novel, compared to published work (for instance the observed drop in exhumation rates and that flux steady-state is only attained using a three-dimensional geometry).

In the following we provide the point-by-point response towards the comments made by the reviewers. Note that the line numbers refer to the lines in pdf-version of the revised manuscript.

*The reviewers' comments are in italics*, **our response is in bold**. Our changes made in the manuscript are highlighted in blue.

**Reply to reviewer#1**

*The manuscript presents new thermochronological data from the Olympic Mountains in Washington State, USA. In addition, the paper presents a quantification of the influx and outflux of material to the mountain range over the last 14 million years, in order to discuss whether this accretionary wedge range is in a flux steady state. The influx of material into the accretionary wedge is based on knowledge from offshore sediment volumes and plate velocities, whereas the outflux is based on an exhumation map from previous thermochronological work in the range.*

*The topic is interesting and the overall finding represents a new scientific contribution. However, I find it peculiar how the authors present newly obtained data, apparently without using them in the following calculation of the denudation pattern/outflux. As it is presented now, the paper appears quite fragmented, and one wonders what is really gained form including the new data. In this sense, the paper could just as well be split into two separate papers. One on the newly obtained thermochron data, and one on the flux in and out of the range.*

**We thank the reviewer first for the time she or he has spent on revising the manuscript, and second for acknowledging that the results from this work are new and contribute to this topic. We understand that it seems counterintuitive to present new thermochronometry data and to not include them in the influx and outflux calculations. However, we still use the results obtained from the new data in our qualitative discussion of flux steady-state, where the modeled exhumation histories demonstrate a strong temporal variation in exhumation (equating to a variation in the outflux). However, as discussed in the paper, we can't use these results for our quantitative assessment. We also hesitated to include all of the new data and published data in a new 3D Pecube model for reasons further outlined below.**

**In the revised manuscript, we will develop the implications of the new data in more detail, so to demonstrate that our findings are stronger with their inclusion (to avoid a splitting into two papers).**

We emphasize that only our new thermochronometry data reveal the hitherto unnoticed decrease in exhumation rates in the introduction (lines 83 – 85) and conclusion (lines 707 –

714). We also highlight our purpose of taking new thermochronometry samples in order to reconstruct exhumation in the methods section (lines 208 – 211).

*Regarding the flux calculations, more justification is needed for the choice of sediment thickness. It seems like the 1.5 km is taken out of the blue. Also related to this, it should be better explained what the gain is from doing both the 2D and 3D approaches. Why not just do the 3D, and test this with the 2D cross sections? Isn't it obvious that a 2D approach is not ideal when a strong spatial gradient exists in exhumation perpendicular to the 2D section?*

**These are good and helpful remarks by the reviewer. Indeed, we did not explain in detail why we used 1.5 km as minimum sediment thickness. A more detailed comment on this is provided below and a thorough elaboration on this will be included in the revised manuscript.**

**This comment highlights that we did not convey clearly enough our approach for performing both a 2D and 3D flux calculation. Our intention with providing both 2D and 3D flux analyses stems from the fact that the previous attempts for estimating the flux balance in the Olympic Mountains were performed in 2D (e.g. Batt et al. 2001). As mentioned by the reviewer, the spatially variable exhumation rates suggest that the**
**outflux in this mountain range is highly variable, depending on the actual position in the mountain range. Therefore, our intention was to perform an analysis with the established 2D setup and to compare this with an analysis based on a 3D geometry and to see whether different outcomes are obtained. As we describe in the manuscript, on long timescales flux steady-state is only achieved using a 3D geometry, whereas the 2D**
**geometry requires unrealistic parameters (or is an oversimplification). We believe, that this is an interesting outcome, given that the previous analysis has been done in 2D.**

**In the revised manuscript, we will explain in more detail our approach in the methods section and discuss in more detail what is gained from performing both 2D and 3D**
**analyses in section 5.2.**

Regarding the concern of the 2D and 3D flux calculations, we rephrased the introduction and try to explain that the previous approach for assessing flux steady-state in the Olympic Mountains was performed in 2D (lines 55 – 61). We highlight that we want to compare results
from both 2D and 3D (lines 303 – 309), and finally emphasize that in our case, contrary to the previous publication flux steady-state is only obtained using a three-dimensional geometry (lines 601 – 607 and 723 – 725), showing that the geometry inherent for flux steady-state analysis is important.

*Introduction: Lines 56-61: you don't need to summarize the conclusions here in the*
*introduction.*

**This seems to be a choice of style. Since the manuscript has not reached a space limit, we prefer to leave the introduction as it is.**

*Methods: 150-153: Would be great to introduce here already what methods are used for the flux calculations.*

**Thank you for pointing this out, we will insert a brief introduction to our used flux calculations at this section of the revised manuscript.**

A brief description of our flux calculations is now given in lines 203 – 205.

*182-: Although I appreciate that a 1D approach can give valuable results, I cannot stop to wonder why the authors did not take the full 3D approach, which is the core purpose of*
*Pecube. To the best of my knowledge, all the authors are doing could be done in Pecube in 3D. Using all existing data, they could make an updated exhumation map for the region to be used in their calculations of the outflux. You should as a minimum discuss why it is not feasible to do a full 3D inversion in Pecube.*

**Indeed, the prime purpose of Pecube is to perform full 3D inversions of**
**thermochronometric datasets. After inspecting our thermochronometric dataset (particularly the samples with multiple thermochronometer systems available), we suspected a possible strong variation in exhumation rates, because of the observed ages (e.g. young AHe ages, overlapping, 5–7 Ma old AFT/ZHe ages). Hence, our prime intention was to evaluate this hypothesis. To perform a full 3D model in Pecube, an**
**exhumation rate pattern, an exhumation history (so temporal variations in exhumation) and also topography need to be prescribed in advance. In the 1D model, thousands of possible exhumation histories are explored with the Monte-Carlo algorithm, yielding a best-fit history for the observed thermochronometer ages for the respective sample. The advantage to this method is that histories with variations in exhumation rates can**
**be robustly constrained (and do not have to be guessed in advance by the modeler in a 3D approach, which could introduce a bias). We use the results obtained by this approach for our qualitative assessment of the flux steady-state, where we argue for temporal variations in the outflux, because also the exhumation rates are temporally variable.**

**As mentioned by the reviewer, we could use all existing thermochronometric data and use the temporal evolution of exhumation rates obtained from the 1D modeling to set up a 3D model. However, the differences in exhumation rates between the seven samples modeled with 1D (e.g. the timing and magnitudes in variation of exhumation rates) would be difficult to account for in a 3D model, and large misfits between**
**modeled and observed ages would be expected, if all existing thermochronometer ages (from this work, as well as literature) are included in a 3D model. Such misfits could be**

expected given that samples are located in different parts of the accretionary wedge, which are probably separated by faults. Setting up a 3D Pecube model with faults on top of the ellipse exhumation rate pattern in order to simulate different parts of the accretionary wedge would be a paper on its own and many additional parameters would need to be constrained. Hence, we did not include a model like this in the current manuscript.

For our quantitative outflux analysis, there would also be no benefit from an updated exhumation map (with an additional temporal variation in exhumation). The integrated amount of exhumation considered during the 14 Myr period (this value is used during the outflux calculations) would still be the same as with the current exhumation rate map (i.e., even a more sophisticated model will still result in the same amount of exhumation).

In the revised manuscript, we will include a more thorough discussion (including the remarks above) about the concerns raised by the reviewer.

We reformulated parts of the methods section on the thermo-kinematic modeling (lines 242 – 290), particularly addressing the poor spatial resolution of the seven samples considered for modeling (lines 280 – 284). In the new limitations section (Section 5.5), we also emphasize that in order to create a new 3D model including all data and considering the observed temporal variations in exhumation would require to constrain parameters like faults, which is not feasible with the data available at present (lines 633 – 644).

*220-236: this section could be clearer and more up-front about the 2D vs. 3D approaches. Why not just use the 3D approach? What is gained from the 2D? this should be made clear.*

See our response to the reviewer's comment above.

*252-255: the use of 1.5 km as the minimum for the previous thickness of offshore sediment is not properly justified. It is stated specifically that this is the largest unknown, and right now it seems you have grabbed this number out of thin air. Could there not have been more sediment earlier where you argue for a much higher exhumation rate?*

Thank you for pointing this out. We agree with the reviewer, that the minimum sediment thickness of 1.5 km needs to be better justified. A more thorough way of constraining this parameter is explained in the following, and we will modify the manuscript to explain this better. More specifically, during the entire history of sediment accretion at the deformation front since ~40 Ma, the sediment thickness was likely different from the reported 1.5 km, but to get constraints on the sediment thickness for this entire period is difficult, as most of it has already been incorporated in the accretionary wedge. However, we focus our analysis on the past 14 Myr, where obtaining constraints is easier. As shown in Figure 3 of the manuscript, the oceanic crust subducted at present is very young (~6–9 Ma). The young age together with the fast subduction rate (which was even faster prior to 6 Ma) prevent the accumulation of a thick succession of sediments on top of the oceanic crust (e.g. the pre-Quaternary sedimentation rates obtained from the ODP boreholes are around 80–110 m/Myr, Table 1). Assuming it takes the oceanic crust about 9 Myr to reach the deformation front (the oldest oceanic crust at the deformation front is currently 9 Ma old) would yield a sediment thickness of ~700–1000 m at the deformation front. However, this is likely an underestimation of the actual thickness, because the inherent assumption for this calculation is that the spreading rate and convergence rate stay constant over time. Furthermore, the sedimentation rate likely increases closer to the deformation front, because more detritus is delivered through submarine canyons and turbidity currents (the ODP boreholes were drilled on the deep sea plain or on a submarine fan ~120 km away from the deformation front, Figure 3). Therefore, using a minimum sediment thickness of 1500 m for pre-Quaternary times should correspond to a good estimation of this otherwise difficult to constrain parameter.

As the reviewer mentions, we observe high exhumation rates at ~6 Ma, which consequently yields a higher sediment supply to the ocean. But the high exhumation rate period is followed by a period of very slow exhumation rates from ~6–2 Ma (Figure 8), so the sediment supply to the ocean will also decrease. Furthermore, the sediment material entering the accretionary wedge at the deformation front is a mixture of material delivered from different sources (e.g. Olympic Mountains, Vancouver Island, Canadian Cordillera). Variations in exhumation rates in these different source regions will also result in variable amounts of sediment from the respective source region, however the effect of these variations on the sediment thickness is difficult to constrain.

A more thorough elaboration on the used sediment thickness will be included in the revised manuscript.

In line 326 – 327 we refer to the revised electronic supplement, which now contains a detailed explanation about the Pre-Quaternary sediment thickness in section S3.1.

*Lines 295-298: as mentioned above, it seems odd that you don't want to actually use the data you present here. Either you should use all available data together, or you can just as well leave the new data out.*

We understand the concern of the reviewer, to not use the presented data in all of our analyses. As outlined in the comments above, we are unfortunately not able to include the data in a new 3D Pecube model, to obtain an updated exhumation pattern (for the quantitative flux calculations). However, we still use information obtained from the new data for qualitative flux analysis, clearly showing variations in exhumation rates. The change in exhumation rates at ~6 Ma due to a change in convergence rate is also a new observation, that has previously not been reported for the Olympic Mountains and we will clearly state this in the revised manuscript.

*Results: 312-318: I don't see AFT ages mentioned here?*

**Thank you for indicating this, the AFT ages will be included in the revised manuscript.**

The AFT ages of the respective samples are now mentioned in lines 402 – 410.

*345: should it not be OP1551 to be consistent with figures?*

**Indeed, this typo will be corrected in the revised manuscript.**

The correct sample ID OP1551 is now given in line 441.

*357: I would argue that the volumes vary with the location in the wedge geometry, latitude is irrelevant.*

**We will consider the suggestion and replace "latitude" in the revised manuscript.**

We rephrased latitude with " … depending on the position within the wedge." (line 454 – 455).

*Discussion: Lines 378-382. This is unclear. You start with: "In the absence of a strong lateral gradient". . . and end with "due to the strong spatial gradient. . ."*

**Our intention was to quickly introduce the classic way of interpreting an age-elevation transect (which requires the absence of a lateral gradient in exhumation). However, this is not possible in the Olympic Mountains (due to the spatial gradient in exhumation rates). We will rephrase this paragraph in the revised version of the manuscript, to make it clearer.**

This entire section of the discussion was rephrased and the particular paragraph the reviewer is referring to is reformulated (lines 477 – 483) so that it should be better understandable now.

*Lines 401-402: references are needed here, or rephrase to avoid passive voice.*

**References to Michel et al. 2018 and Brandon et al. 1998 will be included in the revised manuscript.**

This entire paragraph is now significantly changed, so that we did not have to include references (lines 502 – 511).

*Lines 480-482: more likely a lower outflux outside of the profile, is it not?*

**Thank you for indicating this, we will include this in the revised manuscript.**

**Reply to reviewer#2**

*This paper by Michel et al. aims at assessing the degree of steady-state of the accretionary prism of the Olympic Mountains. The authors used existing and new thermochronological*
*data, to assess by an inverse model, the evolution of exhumation and denudation during the last 14 Myr that they equate to material outflux. For influx, they consider the rate of plate tectonic convergence multiplied by sediment thickness at the wedge front, and it is implicitly assumed that only frontal accretion contributes to material influx (a largely questionable hypothesis).*

*The question addressed by this paper is of interest for a large community, and I thank the authors for the efforts they have put in this manuscript. However:*

**We thank the reviewer for his time she or he spent on reviewing our manuscript and that she/he agrees, that this topic is of interest for a large community.**

*1) The method used (inverse model based on Pecube with only 1D temperature and*
*discretized at 1 Myr) is not optimal.*

**It appears we did not succeed in conveying our main purpose of using this modeling approach. We comment in detail on this concern in our responses below and show how we will address this in the revised manuscript.**

*2) The influx reconstruction is not constrained or discussed well enough, in particular the*
*contribution of fluxes other than frontal accretionary fluxes (such as isostasy, etc) or temporal changes in the structure of the wedge are not inferred.*

**These are interesting remarks that definitely need to be addressed. We provide a more detailed response in the comments below.**

*3) The paper is sometimes confusing (e.g., some thermochronological data are not all used in the inversion, the 2D or 3D accretionary fluxes are not necessary (only in 3D or in 2D)).*

**Thank you for pointing this out. We only used the thermochronometry data from seven samples in our modeling, because these seven samples are multi-chronometer samples, so that three or four thermochronometer systems (AHe, AFT, ZHE, ZFT) are available for each sample. We omitted the remaining samples, because for these only**
**one or two thermochronometer systems are available (e.g. AHe and ZHe) or most of the thermochronometer systems are unreset. Hence the seven considered samples yield the best-constrained exhumation histories. Relying on these specific samples provides the most consistent framework for equal comparison of exhumation histories across the range.**

**Our purpose for providing influx calculations both in 2D and 3D was to tie in with previous calculations of flux steady-state, which were performed in 2D (Batt et al., 2001) and to test whether the assumed spatial geometry is important for the flux steady-state analysis. Indeed, the calculations both in 2D and 3D showed that different results are obtained, depending on the assumed geometry.**

**For the revised manuscript we will explain in more detail the reasons behind our sample selection for the inversion and why it matters, whether the flux analysis is performed in 2D or 3D.**

We now explain in more details the reason behind our sample selection for the thermo-kinematic modelling in lines 246 – 252.

We reformulated the introduction, so that it now includes a short description of the approach of Batt et al. (2001) in lines 55 – 61 and emphasize that we want to explore whether steady state depends on the spatial geometry (lines 81 – 83). Because in our case we obtain steady-state only for a 3D geometry, we now highlight this in the discussion (lines 601 – 607) and conclusion (lines 721 – 725).

*4) The organization of the paper itself could be better to help its readability (for instance there is no results section on steady-state itself, which is surprising having read the title of the paper).*

**In section 4.3 we provide results from our quantitative influx and outflux calculations but limit our explanations to only reporting the numbers. Any further elaboration (e.g.**
**direct comparison between influx and outflux volumes) would already be discussion,**

which can be found in section 5.4. This also pertains to the qualitative comparison between influx and outflux, based on variations on parameters like plate convergence rate, sediment thickness and exhumation rates.

*5) The addition of this paper, compared to established literature (Batt et al., 2001; Brandon et al., 1998; Michel et al., 2018) that already demonstrated a global steady- state over the last 14 Myr and change of exhumation rate at the onset of the glaciations, is not clear to me.*

With our approach of using multi-chronometer samples we are able to report a remarkably rich temporal exhumation history that represents a significant new contribution. Besides the increase at ~2–3 Ma (already reported by Michel et al. 2018) there is a decrease in exhumation rates occurring at ~6 Ma due to the decrease in plate convergence rate. Since the contribution of both tectonics and climate to the evolution of mountain belts is still a matter of active debate in the geoscience community, this result presented in the manuscript is of interest also to a broader readership.

In a previous study, Batt et al. (2001) concluded that the Olympic Mountains are in flux steady-state, because they successfully modeled their observed thermochronometer ages with a thermo-kinematic model (including horizontal velocities). However, a balance between influx and outflux is a basic precondition in deriving the mathematic equations for the model. Therefore, a flux steady-state balance is not directly tested, but deduced from their successful modeling. One might also argue that due to the non-uniqueness of modeling approaches (i.e. different models yield the same answer, depending on the considered parameter space), a direct corroboration of flux steady-state in the Olympic Mountains is still missing.

For our approach we tried to directly calculate both the influx and the outflux independently from each other, considering new constraints for parameters like sediment thickness, plate convergence rate and denudation rates. We also considered the temporal variations in these parameters, which has so far not been done. In addition to Brandon et al. (1998) and Batt et al. (2001) who focused their analyses on the long-term flux steady state over 14 Myr, we show that (at least qualitatively) strong temporal variations in influx and outflux occur on short timescales.

Hence, we believe that the results in this manuscript correspond to new results that complement the established literature and are of interest to a broad readership. As the reviewer also mentions in another comment below, we should more clearly explain how our methodology/approach differs from the established literature. We will do this in the introduction of the revised manuscript.

We significantly reformulated the introduction, so that it should now be more obvious what represents a new contribution. This includes indicating the hitherto unnoticed decrease in exhumation rates (lines 83 – 87), and pointing out how our approach for assessing flux steady-state is different from the approach of Batt et al. (2001). In the conclusion we emphasize that with our approach we only obtain flux steady-state for a three-dimensional geometry, contrary
to the previous analysis (lines 723 – 725).

Detailed comments:

*# L 32-35: A very minor comment: steady-state is time-scale dependent, but it is also spatial-scale-dependent. The likelihood of obtaining a steady-state, for a given time-scale, is likely decreasing when going to finer spatial-scales due to heterogeneities and variabilities in*
*landscape dynamics or tectonics that might become more and more dominant in controlling averaged or integrated values (i.e., mean topography or mountain range sediment discharge). This applies to topographic and flux steady-states. This dual dependency of steady state to time and spatial scales (if correct), implies that defining a steady-state over a time-scale can be a correct or incorrect assumption, depending on the associated spatial-scale. So please be*
*clear and explicit on which spatial-scale you investigate here.*

**Thanks for this important remark. Indeed, as we showed with our flux steady-state analysis (performed both in 2D and 3D), the spatial geometry considered for the analysis is important. We will be clearer in the revised manuscript, which spatial geometries we investigate.**

We reformulated this paragraph of the introduction, addressing now the spatial and timescale dependence of flux steady-state (lines 38 – 44).

*# L 51 - Please add a sentence to clearly state in the introduction, what are the main differences, in terms of methods and expected results, of this new manuscript compared to Michel et al. (2018) at Geology. Their abstract ends with: "However, the youngest AHe ages*
*require a 50–150% increase in exhumation rates in the past 2–3 m.y. This increase in rates is contemporaneous with Pliocene-Pleistocene alpine glaciation of the orogen, indicating that tectonic rock uplift is perturbed by glacial erosion." Having read Michel et al. (2018), the differences are not clear to me at this stage of the paper (without having read the following sections).*

**Michel et al. (2018) focused on presenting results from thermochronometric dating (AHe and ZHe) and thermo-kinematic modeling, documenting the role played by the subduction zone geometry in driving rock uplift and the impact of Plio-Pleistocene**

glaciation (increased exhumation rates due to glacial erosion). A direct test of flux steady-state was not the premise of that paper.

In the present contribution, we capitalize on multi-dating samples (additional AFT and ZFT ages), revealing an additional temporal variation in exhumation rates, which could not have been disclosed by Michel et al. (2018) due to the missing FT data. We also assess the flux steady-state balance by investigating both the influx and the outflux independently from each other.

To address the concerns of the reviewer we will convey more clearly in the revised manuscript what discerns the new contribution from previous work.

We reformulated the entire paragraph (lines 73 – 91), pointing out that the decrease in exhumation rate has not been discovered by Michel et al. (2018) and how our flux analysis is different from the approach of Batt et al. (2001).

*# L 202-207: I don't understand the need to use Pecube if neglecting topography and considering only a 1D model. An inverse modelling strategy using for instance QTQt (Gallagher et al., 2005 and so-on), that can jointly inverse samples from the same vertical profile, seems more appropriate here. This approach also has the benefit of not requiring any a priori time discretization. These advantages prevent the potential for both over interpretation*
*and the introduction of artifacts in the inferred thermal histories.*

It is important to emphasize that models such as HeFTy and QTQt are purely thermal models. These models are designed to be very good in calculating temperature-time paths based on thermochronometry information. However, for the purpose of this study we are exploring kinematic information (e.g., exhumation), and therefore, require a
thermal-kinematic model such as Pecube. Furthermore, as we try to discuss in section 5.1 of the manuscript the information that can be obtained from our age-elevation transects is only limited due to the horizontal extent of the transects and the strong lateral gradient in exhumation rates within the Olympic Mountains. So, although vertical profiles can be directly included in QTQt, this might not yield ambiguous results.

Previous exhumation rates reported by Michel et al. (2018) for the Olympic Mountains were also derived by thermo-kinematic modeling using Pecube (however by using a 3D model). To tie in with previous work, we again used Pecube, but due to the suspected temporal variations in exhumation (as suggested by the thermochronometry data) we used the 1D model option coupled with a Monte-Carlo algorithm. This approach has

**successfully been applied in other studies to derive exhumation rate histories, e.g. Ehlers & Thiede (2013), Adams et al. (2015), Avdievitch et al. (2018).**

**The issue mentioned by the reviewer regarding the timestep discretization is a good remark and requires more explanation. Our detailed response to that point can be found below and will be included in the revised manuscript.**

*# L 230-236: What about isostasy or dynamic topography: are they not considered in the material influx? This need to be discussed.*

**We comment in detail on these points in our response below.**

*# L 249-251: There might be some circularity in the rational (to assess steady-state), as sediment thickness depends on the outflux and controls the influx.*

**We thank the reviewer for pointing this out. This indeed is an aspect that represents a limitation to our approach. As we described in section 2.3 of the manuscript, the sediment currently entering the accretionary wedge is a mixture of sediment with different source regions (e.g. Olympic Mountains, Vancouver Island, Canadian Cordillera and in case of the Astoria fan the interior USA). Particularly with the onset of**
**Plio-Pleistocene glaciation this effect became more pronounced, due to increased detrital input from the Cordilleran Ice Sheet. Hence, our influx/outflux calculations for the Olympic Mountains do not represent a closed system, where increased denudation in the Olympics (so a higher outflux) directly results in a higher influx (due to an increased sediment thickness).**

**However, our calculations indicate that on long timescales (i.e., the 14 Myr considered in the calculations) flux steady-state is reached, which might seem surprising given that the sediment thickness is governed by contributions from different source regions. We suspect that processes during sediment deposition, like redistribution by turbidity currents and re-deposition in more proximal parts of the Juan de Fuca plate, play an**
**important role in the final sediment budget. In other words, the amount of sediment eroded from the Olympic Peninsula in a given time period (the outflux) is dispersed as it enters the ocean, so that for the same time period only a fraction of the sediment thickness (governing the influx) is composed of material originating from the Olympic Peninsula.**

We now included a new section in the discussion (Section 5.5), where we discuss the limitations and restrictions of our approaches. The elaboration on this point mentioned by the refer can be found in lines 655 – 672.

*# L 295: "Our exhumation rates presented in this paper (Fig. 5) have a high temporal resolution". This statement might be overstated. No test was performed on the sensitivity of*
*the inversion scheme and results to the time-step used (only 1 Myr was used). The data, especially individual samples, do not necessarily inform on a temporal evolution at a 1 Myr resolution. Is the inversion misfit better when changing temporal resolution? I would like to see some tests to determine the best time-step for the inversion (at least performed on one sample).*

**Our apologies, if our wording caused some misunderstanding. With this sentence we wanted to state that our 1D model is well suited to resolve temporal variations in exhumation rates, because the model covers a large range of possible exhumation histories. In comparison, the exhumation rates used in calculating the denudational outflux (the rates presented in Michel et al. 2018) have a good spatial coverage, but the**
**3D model is not suited to investigate a large range of possible exhumation histories. In the revised manuscript we will change the wording, so that our intension for using the exhumation rates during the flux calculations is more evident.**

**Nonetheless, the concerns of the reviewer regarding our chosen time-step interval are justified and need to be addressed. We will perform tests for a sample using different**
**time steps. However, the time period resolved by our thermochronometer ages varies between 5 and 10 Myr for the seven considered samples, placing a threshold to possible durations of the time steps (e.g. durations longer than 3 Myr seem not to be reasonable). We also note, that although our reported exhumation rate histories in Figure 5 seem very detailed and suggest a high temporal resolution, these step-like**
**patterns would likely be much smoother in reality. Hence our summary of exhumation rates in Figure 8 uses a more smoothed shape.**

**In the revised manuscript, we will present results from the sensitivity analysis of the time-steps.**

We rephrased the statement the reviewer criticized so that it should now be less misleading
(lines 380 – 383).

As requested, we also conducted additional simulations using different time step intervals and provide the results along with a brief discussion in the electronic supplement (in Section S2). We direct the reader to the supplement in lines 255 – 257 of the revised manuscript.

*# L 510 – section Results - Please add a sub-section at the end of the Results section, to*
*present the results concerning flux steady-state analysis. This is the main ambition of this paper, and yet there is no result section on steady-state. This does not help the reader to get a clear message from reading this paper.*

**See our response to comment 4).**

*# L 367 – section Discussion - Please add a sub-section to present the limitations of the*
*approaches used in this paper.*

**A good suggestion by the reviewer. We will include a section in the revised paper, which addresses the aspects mentioned by referee #1 and #2 (e.g. limitations of our influx and outflux calculations, limitations of the exhumation rates).**

As requested, we provide a new section (Section 5.5) discussing the limitations and
restrictions of our approaches (lines 630 – 706).

*# L 385-390: For the vertical profile with ZHe data, the change of polarity of the slope is not a robust feature. Models can be defined, satisfying all the ZHe ages and their uncertainties, without leading to a change of slope. However, for AHe, the change of slope seems robust.*

**We agree with the reviewer that a line can also fit the ZHe ages (without a change in**
**slope). In the revised manuscript, we will change this. Our further interpretation that the age elevation transects only yield limited information on exhumation rates should still be valid.**

We changed Figure 4 in order to account for the comment made by the reviewer, and also reformulated the entire section about the age-elevation transects in the revised manuscript
(lines 477 – 500).

*# L 451: It is assumed in this paper that the geometry of the accretionary wedge is constant and that other processes than tectonic accretion are negligible (an implicit assumption). Could you please discuss: 1) if there were some potential changes in the extent, volume and geometry of the Olympic Mountain accretionary wedge? 2) how you integrate isostasy or*
*dynamic topography in your comparison of in- and out- fluxes? The isostatic response to*

*erosion can generate uplift (with no associated influx in the presented model) and induce additional erosion. This need to be discussed in this manuscript (not as a perspective L533-536).*

**We thank the reviewer for raising this issue and we will discuss it in more detail. For our quantitative assessment of flux steady-state we try to calculate the actual volumes of influx and outflux during the 14 Myr period. The two calculated fluxes should be viewed as pure volumes and separately from each other. Flux steady-state is attained, if both volumes (calculated for the 14 Myr period) equal themselves. We base our outflux calculations on the spatial integration of the exhumation rate pattern presented by Michel et al. (2018) and also consider the increase in exhumation due to the Plio-Pleistocene glaciation. As influx we consider all sediment resting on the subducting oceanic plate (so the influx is governed by the sediment thickness and the plate convergence rate). We assume that the mechanism of actual accretion (i.e., frontal accretion vs. sedimentary underplating) is not important, because finally all sediment is incorporated into the accretionary wedge and contributes to the influx volume, irrespective whether accretion takes place at the front of the accretionary wedge or at depth. As we discuss in the manuscript there are no estimates available on how much sediment might be transported into the mantle, so this is a limitation to our approach.**

**The impact of isostasy to the influx volume is not obvious to us. Any isostatic response (e.g. due to increased erosion during the Quaternary) affects the mountain range itself (so to say the volume of already accreted sediment material), but does not directly affect the influx volume. A possible increase in erosion related to isostatic uplift of course affects the outflux, as the reviewer noted (and in turn increases the influx, due to a higher sediment yield). If there was a significant change in rock uplift rates created by a change in isostatic compensation, then this would have likely changed exhumation rates, which would be recorded by our chronometers and seen in our modeling results. Therefore, there is no reason to directly account for isostatic compensation in our thermo-kinematic models.**

**Effects of the Plio-Pleistocene glaciation are already included in our calculations (see Table 5), like an increase in sediment thickness (affecting the influx) and an increase in denudation rates (affecting the outflux). So one could argue, that a possible contribution of the effects of isostatic uplift is already included in our calculations (but we can't quantify the exact contribution of isostasy). In total, the observable effects of Plio-Pleistocene glaciation are likely the combined effects of increased sediment supply (particularly by the Cordilleran Ice Sheet), and increased exhumation rates (due to glacial erosion).**

The reviewer also asks, if there were some potential changes in the extent, volume and geometry of the accretionary wedge. These parameters are affected by the amount of accreted sediment and the dip of the subducted plate. As shown in Figure 1 the
subducted plate displays a bend below the Olympic Mountains, resulting in a flatter angle of subduction compared to areas in the north or south. If bending the plate occurs through time (the reason for bending is hypothesized to be extension in the Basin and Range Province, Brandon & Calderwood 1990), then this indicates that the angle of subduction below the Olympic Mountains is likely variable through time and
hence the volume and extent of the accretionary wedge is varying through time as well. Our reported cross sections (Figure 7) show that the volume and shape of the accretionary wedge is spatially variable. As stated in our manuscript, a change in the width and mechanics of the offshore part of the wedge has been observed due to increased sedimentation during the Quaternary (e.g. Adam et al., 2004). So, the
temporal and spatial evolution of the accretionary wedge is complicated. However, our flux steady-state calculations should be independent from any changes in the shape of the accretionary wedge. As we outlined above, we only look at the calculated volumes of influx and outflux, which do not depend on any parameter affecting the shape of the accretionary wedge.

For the revised manuscript, we will include a discussion of the above points, likely in a section about the limitations of our approach, as also suggested by the reviewer.

As suggested by the reviewer, we included a new section (section 5.5) in the revised manuscript, where we discuss the points raised by the reviewer (lines 630 – 706).

*# L 505: "In summary, the assessment of flux steady state in the Olympic Mountains is non-*
*trivial and many scenarios are possible." The used datasets (thermochronological data, sediment deposits, geometrical structure, etc) are not sufficiently well resolved to offer a robust assessment of temporal changes in fluxes or in steady-state. Therefore, one could question the real addition of this paper compared to Batt et al. (2001), Brandon et al. (1998) or Michel et al. (2018) that have already demonstrated 1) a global steady-state over the last 14*
*Myr and 2) a potential change in exhumation rate with the onset of Plio-Pleistocene glaciations.*

In our response to the reviewer's comment 5) we already outlined what we view as new contributions. We fully understand that the reviewer is critical on this point. The Olympic Mountains have long been viewed as a case example for a (flux) steady-state
mountain range. As we tried to convey in our manuscript our approach shows that both influx and outflux are subject to temporal variations (but as the reviewer notes, we can

**not provide a full, quantitative assessment on small temporal scales). We believe that the results reported in this paper will stimulate further investigations of (flux) steady-state, so that in the future better and further constraints are available for parameters, which we could not include in detail (e.g. sediment data, margin parallel velocities). This could contribute to a new perspective and understanding of steady-state in active mountain ranges.**

We reformulated parts of the discussion, so that section 5.4.2 now points out the difference between our approach and the approach of Batt et al. (2001), because we only observe flux steady-state for a three-dimensional geometry (lines 602 – 607). The rewritten conclusion is now also more specific about what new contributions were made with this manuscript (lines 707 – 733).

*Minor edits:*
*# L 14 : "We present 61 new thermochronometric ages" - Please add: mainly obtained from 21 new samples (or the correct number).*

**As requested, we will change this in the revised manuscript.**

We added the correct number of new samples (line 15).

*# L 37: "tectonic parameters" – please change by "tectonic conditions" (a parameter implies a quantitative framework/model that has not been defined yet).*

**Good remark, will be changed in the revised manuscript.**

We replaced the word "parameter" with condition at this location and also throughout the entire manuscript.

*# L 39: "Plio-Pleistocene glaciation" - There is probably no need to limit the scope to the onset of Plio-Pleistocene glaciation, as older glaciations (for instance at the Eocene- Oligocene transition; Bernard et al., 2016; Thomson et al., 2013) might have also led to variations in denudation and exhumation.*

**In the revised manuscript, we will no longer limit this to the Plio-Pleistocene glaciation and include references to earlier glaciations.**

We include the two additional references mentioned by the reviewer, and no longer limit the effects of glaciation to the Plio-Pleistocene glaciation (lines 49 – 52).

*# L97: ref "Ehlers et al. 2005" : The closure temperature of these thermochonometers has been constrained in older papers than Ehlers, 2005. For instance: Gallagher et al., 1998; Farley, 2002; ...*

**Additional references will be included in the revised manuscript.**

We included an appropriate reference for each thermochronometer system (line 133).

*# L 162: "three/two" - What does three/two mean here?*

**This indicates the number of samples collected for this study, which have been dated with AFT and ZFT, respectively. We will reword this in the revised manuscript.**

The reworded phrase, which is now more explicit, can be found in lines 221 – 222.

**Marked-up version of revised manuscript:**

[revised manuscript text omitted]

---

## Author Response (AR2)

**Authors' response to reviews suggesting minor revision**

For Manuscript esurf-2018-65: "How steady are steady-state mountain belts? - a re-examination of the Olympic Mountains (Washington State, USA)"

By: Michel et al.

Dear Editor,

we want to acknowledge the time and effort the two reviewers as well as the associate handling editor have spent in reviewing and handling our manuscript also during this second round of review. We are delighted that the changes we made to the first version of the manuscript meet with the reviewers' approval. The comments and suggestions the reviewers are providing during this second round of review are again very thoughtful and productive.

Based on these we made same minor changes to the manuscript, which do not conflict with the main story of the manuscript. Reviewer #2's issue with Equation 2 was, indeed, a typo, so the calculations provided in the paper are correct. We hope our response towards the comments by the reviewers and the changes made to the manuscript are sufficient and meet the editor's expectations.

In the following, we provide a point-to-point response to the comments by the reviewers. At the end, the manuscript can be found in a track-changes style, which allows to easily identify changes we made.

On behalf of the authors,

Lorenz Michel

Below you find our response to the reviewers' comments. The number lines refer to the new version of the manuscript.

*Original reviewer comments are in italics.* **Our response is in bold** and what we changed in this version of the manuscript is colored blue.

**Response to comments from Reviewer #1:**

*The authors have in general responded well to all the comments raised by the reviewers. However, there is a tendency that the authors try to explain their way out of the comments instead of making changes to the manuscript. For instance, both reviewers have argued that they find the 2D flux calculation needless. The fix for these comments has been to elaborate even more on the text, which just makes the manuscript longer and protracted. In the interest of making the paper clearer, more transparent and more readable, the authors may want the change this strategy.*

We thank the reviewer for acknowledging our responses towards the comments by the reviewers during the first round. Again we like to acknowledge for the time she or he has spent during this second round of revision. Regarding the issue of the 2D flux calculations we will provide a compromise and move the 2D calculations and discussion on these to an appendix. See also the more detailed comment below.

*Below are a few comments that remained while reading the revised text.*

*1. Perhaps my comment was not clear earlier, regarding the use of Pecube in 3D. My suggestion was to use Pecube in 3D in inversion mode. However, I agree with the authors that this could be a next step, and something for the future. But such approach would allow you to include all available data, also the ones with only one thermochron, all together. An inversion like that would not be able to fit all data, but would find the best suitable model on the basis of all data.*

Simulations using Pecube in 3D, either in inversion or forward mode, will always provide a sufficient fit between modeled ages and observed data. However, the robustness of a suggested cooling history from a 3D model depends to a large extent on the assumed boundary conditions like exhumation history, topographic evolution, thermal boundaries, and the kinematics. As we showed, the history and evolution of the Olympic Mountains is very complex, and several temporal variations in the described boundary conditions seem likely. This is where we currently see the possibility for over-interpreting the results from 3D simulations without more independent constraints on the boundary conditions. Our simple, 1D simulations provide a robust constraint on the exhumation history of a particular sample (but for example do not include the effects of topography). So the series of simulations should provide information on the exhumation within the Olympic Mountains at the respective sample location. That might be a starting point, where more sophisticated 3D models could tie in. As the reviewer indicates, we believe this is future work, which should include a thorough assessment of all the used boundary conditions.

*2. Also related to the thermos-kinematic modeling, the new comment on time-step sensitivity is not very clear (new manuscript lines 255-257). If there is nothing to gain from the sensitivity analysis I don't see the need to include it. However, that being said, I am not sure that the original comment by reviewer 2 has been adequately met, since the authors only have tested a larger timestep and not a smaller one. But perhaps reviewer 2 has commented on this as well – the original comment by reviewer 2 on this could have been clearer.*

**Based on the comment by the reviewer, we included an additional simulation using a 0.5 Myr time step in the supplementary material. As we explain there, all investigated time steps (0.5 Myr, 1 Myr, 2 Myr, 3 Myr and a time step controlled by thermochronometer age) yield a similar pattern of exhumation for the considered sample. This is a clear indication, that the modeling is not sensitive to the time step. A time step shorter than 1 Myr results in exhumation rate "spikes" that are not resolvable with our thermochronometer data. Hence, using a 0.5 Myr time step would result in an over-interpretation of our data. Due to this, and other reasons outlined in the supplement, we use a time step of 1 Myr for our modeling effort.**

In the revised version of the manuscript we changed the corresponding lines, so that it should now be more evident to the reader (Lines 250 – 254): "We performed a sensitivity analysis in order to find the most suitable time step for our simulations and the results of that analysis can be found in the supplementary material (Section S2). Based on the analysis, a time step interval of 1 Myr seems to be most appropriate to use, given the range of our thermochronometry ages and their respective uncertainties."

*3. A minor comment regarding the flux calculations. I appreciate the efforts the authors have made to elaborate on this. However, I still feel that the authors could be a bit more specific, and simply mention explicitly that "the variable with the greatest uncertainty is the sediment thickness" is actually "the variable with the greatest uncertainty is the sediment thickness back in time, that has now been subducted below the Olympic Mountains". Since the authors mention that this is the variable with the largest uncertainty, I would think that they would include something on this in their new section 5.5 (on the actual numbers used for sediment thickness).*

**Thank you for again highlighting this aspect of our calculation and for bringing up this thoughtful sentence. We followed the suggestion by the reviewer and included a short discussion on the uncertainties of the sediment thickness in Section 5.5, indicating that the reported sediment thickness has uncertainties and could impact the calculated influx volumes.**

We changed the wording of the sentence the reviewer is referring to, which now reads as "… the variable with the greatest uncertainty is the sediment thickness back in time, that has now been subducted below the Olympic Mountains" (Line 322 – 323).

In section 5.5 we included the following paragraph (Lines 625 – 639):

"As we mentioned in Section 3.3.1, a variable with great uncertainty is the sediment thickness over time, which has now been subducted below the Olympic Mountains. In the supplementary material (Section S3.1) we outlined our approach for assessing the pre-Quaternary sediment thickness, which is

used in our calculations. Although the reported 1.5 km sediment thickness seems to be a plausible value, we note that this value is afflicted with uncertainties and might have been higher. Nonetheless, our proposed balance between influx and outflux might still be tenable, even if the pre-Quaternary sediment thickness deviated from the assumed 1.5 km. I.e., we suggested an influx volume of 75–78 x $10^3$ km$^3$ and calculated outflux volumes between 75 x $10^3$ km$^3$ and 82 x $10^3$ km$^3$ (Table 5), so even an additional influx volume due to a thicker, unnoticed sediment thickness could be balanced with our calculated outflux volumes. Another simplification in our calculation is the assumption of a spatially uniform sediment thickness over the considered length. Figure 3 shows that the sediment thickness along the deformation front is variable and is highest in the Nitinat and Astoria fans. However, an attempt to reconstruct along-strike variations in sediment thickness over time is challenging and would introduce further uncertainties, and thus, we assume an average, constant thickness."

*4. It is a matter of taste, and I will leave it up to the authors to decide on this, but I just want to stress once again that you don't need to include both the 2D and 3D for influx/outflux in order to argue that the 3D is different/better. Of course 3D is different and better when there are large spatial variations in the pattern you are integrating. This is not "a new and unexpected observation" as you phrase it in your reply. Also, it is not necessary to do both 2D and 3D to link with previous work (especially since you do not discuss the actual difference between your 2D and the previous work). Your comparison of 2D/3D ends up being obvious and in fact dispensable. In the interest of making your paper clear and transparent, I would still suggest to leave it out. You can still argue that the 3D is better, just by referring to previous work.*

**The reviewer seems to be very concerned on this point and the specified advantages in terms of paper clarity and transparency are convincing to us. Hence, we tried to provide a compromise and removed the 2D calculations from the main manuscript to a short appendix at the end of the main text body. So an interested reader can still see our 2D calculations. The discussion in the main text focuses on the 3D calculations and only briefly mentions the 2D calculations and how they differ from the 3D calculations. We hope, this is in accordance with what the reviewer expects, and it definitely aids with clarifying the manuscript.**

Most changes to the text regarding this point were made to section 5.4.2, Lines 580 – 585 contain the short comparison between 2D and 3D.

Because we now explicitly only consider a 3D geometry, we also modified Equation 2, which now contains the length l of the considered area. Furthermore Figure 6 was modified, where the initial subpanel e) was removed (it contained the 2D integrals of exhumation rates).

**Comment Reviewer #2, (Phillipe Steer)**

*Dear authors,*

*I appreciate the amount of work you did to respond in a fair and correct manner to the comments raised by reviewer 1 and myself. This has helped increase the quality of the manuscript. Thanks for that. I therefore see no a priori major reason to preclude the publication of this interesting manuscript.*

**We are delighted that the reviewer is in favor of our replies to the first round of comments and the changes we made to the manuscript. Again, we thank the reviewer for the time he has spent on our resubmitted manuscript and for the comments and suggestion he is providing in this final round of revision.**

*Minor Comments:*

*1) I think there is a mistake in Equation 2. I believe this equation < Vsed=nu.d.t.u > is incorrect (with nu the porosity), as you are here estimating the volume of pores entering the wedge. This equation should be < Vsed=(1-nu).d.t.u > so that you compute the volume of rocks accreted in the wedge. I believe there are three possibilities:*
*- I am wrong.*
*- This is simply a typo, and the correct equation was used in the model, then all is fine. This is what I believe to be the most likely option, and in turn, this is why I consider this is simply a minor comment.*
*- This is not simply a typo, and your model also solves for the wrong equation. Then, this is a major issue with large implications, as this means you have underestimated the accretionary flux by a factor ~2 (nu=0.31) to ~4 (nu=0.22). This would completely change your results and their interpretation.*
*I am sorry I have not seen that during the first round of review.*

**Our apologies, if this created any confusion and thank you for indicating this flaw. The equation contains indeed a typo and the calculations were done with the correct equation (1-porosity), so the values reported in the manuscript are correct.**

We now inserted the correct equation at the respective location in the manuscript (Line 315)

*2) It is not clear to me if you correct for porosity (of exhumed and eroded rocks) when assessing volumes of denudation? If not, there is a consistency issue. Indeed, you should either consider a bulk volume, in both the accretionary and denudation fluxes (porosity is therefore not needed), or a rock volume (without pores), and in this case you need to multiply the*

*denudation by (1-nu), similarly to the accretionary flux. Moreover, the best would be probably to consider a mass budget with fluxes given in mass per time, as the density of accreted sedimentary rocks (without pores) is probably different than the density of eroded rocks (still without pores).*

**Thank you for indicating this inconsistency. Actually, in the previous version of the manuscript we corrected the volumes calculated within the sediment cross sections (Figure 7), but did not correct the denuded volumes. Now we follow the suggestion of the reviewer and also apply the porosity of 6% to the denuded volumes. This value is based on work from Davis and Hyndman (1989), who investigated sediment within the accretionary wedge offshore Vancouver Island and reported values of 4 – 10% for the porosity. We believe that an average porosity of 6% should yield a reasonable value for our calculations. However, this value is much smaller compared to the porosities of the incoming sediment stack (20 – 30 %), which is probably related to the reduction in pore space during sediment compaction and dehydration processes within the accretionary wedge. In the current, revised version of the manuscript all calculated volumes (influx, outflux, cross sections) are corrected for porosity and should yield a correct volume for the accreted or denuded volumes.**

**After applying the porosity to the denuded volumes, we even observe a better match between our calculated influx and outflux volumes, compared to the previous manuscript.**

We provide a short paragraph on that issue in the manuscript within the section of the cross sections (Lines 369 – 372) and the outflux calculations (Lines 393 – 395).

*Very minor comments:*

*Lines 699-703: These two sentences suggest that Batt et al. (2001) and Brandon et al. (1998) use a circular argument (assuming steady-state to ... interpret steady-state). Please rephrase as this is probably incorrect. For instance, in Batt et al. (2201) a convergence rate of 36 mm/yr is assumed based on geological observations.*

**Indeed, this is a circular argument, but this is the way how Batt et al. (2001) designed their flux analysis. They assumed flux steady-state as a null hypothesis and derived the kinematics of their thermo-kinematic model (the horizontal and vertical velocity component within their model domain) from the assumed balance between influx and outflux. They conclude that the mountain range is in flux steady-state, because their model can successfully predict the observed thermochronometer ages. So neither the influx nor the outflux are actually calculated and directly compared with each other. Furthermore, variations in parameters like convergence rate, sediment thickness or exhumation rate have not been**

**considered in their approach. Hence, our strategy differs and we calculate the influx and outflux independently from each other.**

*Equation 2: It is not really satisfying to use the word "volume" for Vsed as Vsed as a unit of [km2]. This is also annoying later in the text when you compare 2D and 3D "volumes". Maybe the easier would be to multiply this equation by a unit length (in the 2D case) to make a 3D column, and by the wedge length in the 3D case (this is done later in the manuscript anyway)*

**Thank you for pointing this out. Because we now focus on the 3D volumes for influx and outflux (related to the comments by reviewer #1), we modified Equation 2, so that it now also includes the length l and the calculated values have units of km$^3$. We removed some parts of the two-dimensional calculations to an appendix at the end of the manuscript. There, we implemented the comment of the reviewer and introduce a unit width of 1 km to yield units of km$^3$.**

*Equation 3: Please, precise the unit of z is [km] and not [m], if I am correct*

**We are not entirely sure, which units the reviewer is referring to. In Equation 3, z is inserted using units of km. In the original publication of Yuan et al. (1994) the equation has the shape of por = 0.6 x exp(-z/L). L is a constant and in our case we applied L = 1.0 km (because our considered sediments represent sediment before they are incorporated in the accretionary wedge). So z must also have units of km. Note however, that the porosity decreases exponentially with depth within the sediment stack of the considered thickness. For that reason we calculated mean porosities for the respective sediment thickness, which are reported in the text (31% and 22%, respectively).**

*Lines 1253-1255 - I would have said exactly the opposite. Generally, we expect the influx (or uplift) to set the outflux (or erosion) at steady-state. This is for instance what occurs in numerical models of landscape evolution. I am not aware of papers clearly demonstrating the opposite - that outflux sets the influx (but maybe this illustrates my lack of knowledge about sediment recycling in accretionary wedges)*

**We fully agree with the reviewer that generally rock uplift exerts a strong control on denudation, and it seems to be more intuitive to assume that the influx controls the outflux. Unfortunately, we formulated the part of the text the reviewer is referring to in a misleading way. Our intention was to raise the point of the origin of the accreted material (so the influx), and not to discuss the mutual control of both fluxes. If the accreted material is assumed to be recycled, such that for a given time the same amount of material denuded from an accretionary wedge (the outflux) might again be accreted to the wedge (the influx), then the**

**wedge behaves as a closed system and has to be in flux steady-state (because both the influx and outflux volumes are identical). However, as we indicated in the manuscript the origin of sediment accreted in the Olympic Mountains is from diverse source regions, in particular since the onset of Plio-Pleistocene glaciation.**

In this revised version of the manuscript we reformulated the part, so that it should be less confusing. It now reads as (Lines 649 – 653):

[revised manuscript text omitted]